# ReSIP: Reinforcement Learning with Symbolic Inductive Planning for Interpretable and Generalizable Pixel-Based Control

## Abstract

Deep Reinforcement Learning (DRL) has struggled with pixel-based controlling tasks that have long sequences and logical dependencies. Methods using structured representations have shown promise in generalizing to different objects in manipulation tasks. However, they lack the ability to segment and reuse atomic skills. Neuro-symbolic RL excels in handling long sequential decomposable tasks yet heavily relies on expert-designed predicates. To address these challenges, we propose ReSIP, a novel framework for pixel-based control that combines Reinforcement Learning with Symbolic Inductive Planning. Our approach first automatically discovers and learns atomic skills through experiences in simple environments without human intervention. Then, we employ a genetic algorithm to enhance these atomic skills with symbolic interpretations. Therefore, we convert the complex controlling problem into a planning problem. Taking advantage of symbolic planning and object-centric skills, our model is inherently interpretable and provides compositional generalizability. The results of the experiments show that our method demonstrates superior performance in long-horizon sequential tasks and complex object manipulation.

## 1 Introduction

Deep Reinforcement Learning (DRL) has been successfully applied to various fields, including video games (Mnih, 2013), autonomous driving (Sallab et al., 2017), and robotics (Kober et al., 2013). However, building flexible and adaptive robotic agents that can accomplish a diverse set of tasks in novel and complex environments remains a significant challenge in DRL. Such tasks typically demand the agent to formulate long-term plans for logically dependent goals, requiring it to combine diverse skills in complex scenarios involving multiple objects. A significant challenge of these tasks is the need for *compositional generalization*. We can assess it in terms of two distinct factors: (1) different attributes of objects than in training, and (2) different compositions of goals and their corresponding skills, including variations in logical order (Lin et al., 2023).

To address the above challenge, several methods (Zadaianchuk et al., 2020; 2022; Mambelli et al., 2022; Haramati et al., 2024) incorporate structured representations into the DRL algorithms of decision transformers through object-centric representations (OCR). With a powerfully structured representation, they show certain generalizability on the types and numbers of objects in object manipulation tasks. However, they cannot simultaneously learn diverse skills due to the catastrophic forgetting problem (McCloskey & Cohen, 1989), where the new information can distort the previously learned knowledge. Besides, they cannot segment the learned integrated policy into diverse atomic units and reform them to achieve new objectives.

On the other hand, some researchers suggest neuro-symbolic approaches that combine planning and DRL. These approaches aim to handle the combinatorial explosion of possible action sequences by providing high-level abstraction and compositing learned skills. Many existing methods (Illanes et al., 2020; Sun et al., 2020; Zhuo et al., 2021; Mao et al., 2023; Silver et al., 2023) employ a top-down structure by specifying symbolic representation for high-level action models and using them to guide the learning of low-level policies. However, these methods can only work with fully observable

environment states and carefully hand-engineered predicates. These predefined predicates hinder the agent's flexibility, thereby restricting its applicability to complex tasks such as object manipulation.

In this paper, we propose ReSIP, a novel framework for pixel-based control that combines the idea of Reinforcement Learning with Symbolic Inductive Planning.[1] ReSIP is capable of forming a plan that is composed of skills for complex tasks. It enables the agent to learn atomic skills from scratch by exploring simple environments through DRL algorithms (Li, 2017) without relying much on expert knowledge. Furthermore, ReSIP uses genetic programming (Ahvanooey et al., 2019) to induce *symbolic interpretation* for agents' learned skills, including their preconditions and effects. These interpretations provide the agent with a series of fundamental understandings of its learned skills, which are critical for effective planning. During the inference, given a novel and composite task, our agent decomposes the task based on its understanding of the task and atomic skills, formulating a ground skill plan by search algorithms (Abualigah et al., 2021). Then, the agent executes this plan and uses its skills to generate specific actions to achieve the final goal. We experimentally verify the efficiency and effectiveness of ReSIP in two domains: Minecraft, a 2D grid-world environment (Andreas et al., 2017) that focuses on long-horizon planning, and IsaacGym (Makoviychuk et al., 2021), a simulated tabletop robotic environment that evaluates the agent's capacity to manage complex 3D object manipulation. Experimental results show that ReSIP can schedule the sequence of skills in an appropriate order with symbolic interpretation. Moreover, the flexible combination of skills allows our approach to handle environments with varying object attributes.

We summarize our key contributions below:
• **Automatic Skill Discovery.** Compared to previous work, our approach can automatically discover and learn basic skills from the environment without any guidance of designed high-level symbolic representations in advance, reducing the dependency on expert knowledge.

• **Symbolic Interpretation.** We assign symbolic meanings to the learned skills by performing symbolic regression on features. Based on each skill's preconditions and effects, our model can infer the specific task of each skill, thus having a comprehensive understanding of the planning and alleviating the curse of dimensionality. Additionally, this approach is inherently interpretable by planning with a sequential symbolic plan composed of learned skills.

• **End-to-End Pixel-Based Controlling Pipeline.** We propose an end-to-end pixel-based planning framework that can learn skills from scratch and form plans with skill combinations. The final plan we generate is also interpretable.

## 2 PROBLEM FORMULATION

To enable robotic agents to achieve novel, long-horizon goals in multi-object environments, we leverage demonstrations collected in simpler single-object settings and transfer knowledge across tasks via symbolic abstraction. We formalize this setting as a **Goal-Augmented Markov Decision Process (GAMDP)**, where each state is paired with an explicit goal specification. While deep reinforcement learning (DRL) can operate within this framework, it struggles with sparse rewards over long horizons. In contrast, symbolic planners excel in such settings due to temporal abstraction. To bridge these paradigms, we introduce the concept of **skill**: a neural policy annotated with symbolic preconditions and effects, enabling seamless integration of GAMDP-based learning and planning.

### 2.1 GOAL-AUGMENTED MDP

We start from a goal-augmented MDP $\langle \mathcal{S}, \mathcal{G}, \mathcal{A}, \mathcal{P}, \mathcal{R}, \gamma \rangle$ (Liu et al., 2022), where $\mathcal{S}$ is the set of state $s$, $\mathcal{G}$ is the set of goal specification $g$, $\mathcal{A}$ is the set of actions $a$ that the agent executes to interact with the environment, $\mathcal{P}$ is the environmental transition model $\mathcal{P} : \mathcal{S} \times \mathcal{A} \to \mathcal{S}$, $\mathcal{R}$ is defined as the set of reward $r(s_t, a_t)$, and $\gamma \in (0, 1]$ is the discount factor for future rewards. Since the task involves manipulating multiple objects, it is natural to decompose the task into separate goals for each object. An *object* $o \in \mathcal{O}$ has a type, denoted $\lambda_o \in \Lambda$.

---

[1]Code address: https://anonymous.4open.science/r/neurips-fstp-E50B

## 2.2 FROM MDP TO PLANNING

To make long-horizon planning tractable, we first aggregate MDP states into abstract states called features, under the assumption that variations in these features capture high-level state changes. With this abstraction, each skill is annotated with feature-based preconditions and effects. These annotations allow a planning algorithm to efficiently search for a valid sequence of actions.

**Definition 2.1** (Feature). We define the feature $\boldsymbol{f} \in \mathbb{R}^n$ as a vector to characterize the key attributes of the environment. These attributes can either be the object entities, such as `position`, `color`, `shape`, or be the information of tasks such as `reached goal`, `is pressed`. $n$ is the predefined dimension of the feature vector. We define $\mathcal{F}$ as the set of features $f$. We denote the aggregation function as $T_f : \mathcal{S} \to \mathcal{F}$.

We formally defined the mathematical form of the skill based on previous work (Kokel et al., 2021). We group similar operations, forming a skill to address similar tasks. The skill exhibits three key attributes: (1) it serves as the fundamental operational unit for planning, representing a series of actions to achieve a specific goal; (2) it is endowed with a logical structure comprising preconditions and effects; and (3) it demonstrates adaptability by generating specific control actions based on varying input states.

**Definition 2.2** (Skill). We define the skill as a tuple $l = \langle args, \pi, pre, eff \rangle$. Arguments $args \subseteq \Lambda$ is a set of types, specifying the object types to which the skill is applicable. Precondition $pre : \mathcal{F} \times \mathcal{O} \to \{0, 1\}$ is a function that evaluates whether the skill $l$ can be executed on an object $o$ given the current feature $\boldsymbol{f}$. Specifically, $pre(\boldsymbol{f}, o) = 1$ if all conditions are satisfied; otherwise, $pre(\boldsymbol{f}, o) = 0$. Effect $eff : \mathcal{F} \times \mathcal{O} \to \mathcal{F}$ is a function that computes the updated feature $\boldsymbol{f}'$ after applying skill $l$ to object $o$. The goal-conditioned policy $\pi : \mathcal{S} \times \mathcal{S} \times \mathcal{O} \to \mathcal{A}$ maps the current state $\boldsymbol{s}$, a goal $\boldsymbol{g}$, and an object $o$ to an action $a$.

We define the ground skill, denoted as $l(o)$, by substituting the specific object into the policy, preconditions, and effects of the skill. For instance, consider a skill `make_stick` with $args = \{\lambda_{workbench}\}$, indicating that it is applicable only to objects of workbench type. By grounding this skill to a specific workbench $workbench_1$, we obtain `make_stick`$(workbench_1)$, where the preconditions and effects are defined as $pre = \texttt{AtWorkbench}(workbench_1) \wedge (\boldsymbol{f}_{wood} > 1)$ and $eff = \{\boldsymbol{f}_{wood} - 1, \boldsymbol{f}_{stick} + 1\}$. Here, $\boldsymbol{f}_{wood}$ and $\boldsymbol{f}_{stick}$ are elements of the feature $\boldsymbol{f}$, representing the quantities of wood and sticks, respectively. Arguments $args$ and precondition $pre$ can be empty.

For a given goal-augmented MDP, whose initial state is $\boldsymbol{s_0}$ and the goal state is $\boldsymbol{g}$, its features can be written as $\boldsymbol{f}_0 = T_f(\boldsymbol{s_0})$ and $\boldsymbol{f}_g = T_f(\boldsymbol{g})$ respectively. Then, we can form a ground skill plan $\Pi = \boldsymbol{f}_0 \xrightarrow{l_0} \boldsymbol{f}_1 \xrightarrow{l_1} \dots \xrightarrow{l_{n-1}} \boldsymbol{f}_g$ using search algorithms. Finally, we can obtain the trace $\tau = \boldsymbol{s}_0 \xrightarrow{\boldsymbol{a}_0} \boldsymbol{s}_1 \xrightarrow{\boldsymbol{a}_1} \dots \xrightarrow{\boldsymbol{a}_{(n-1)\times t}} \boldsymbol{g}$ by executing each skill's policy $\pi$ in the ground skill plan $\Pi$.

## 3 METHOD

Our goal is to design a framework that can automatically discover and learn atomic skills and form a symbolic plan composed of these skills for complex tasks. The overall structure of our framework is depicted in Figure 1. It mainly consists of two parts: Neuro-Symbolic Skill Training and End-to-End Plan Inference and Execution. We will elaborate on these components below.

### 3.1 FEATURE EXTRACTION

The feature extraction module is the basic component of our framework. It aims to extract compact and disentangled features from raw image states, capturing most of the essential information. Our feature extraction module consists of two layers, which are denoted as $T_e$ and $T_f$. The first layer $T_e$ transforms the input images into object-centric entities, and the second layer $T_f$ further aggregates these entities to form features.

**Object-Centric Representation.** Given a raw image state $\boldsymbol{s}$, we first process it with layer $T_e$, implemented by a pre-trained Deep Latent Particles (DLP) (Daniel & Tamar, 2022). $T_e$ extracts the object-centric representation $\boldsymbol{e} = T_e(\boldsymbol{s}) \in \mathbb{R}^{m \times k}$, where $m$ is the number of objects that appeared in the image, $k$ is the number of object entities, such as `position`, `color`, `shape`. With the

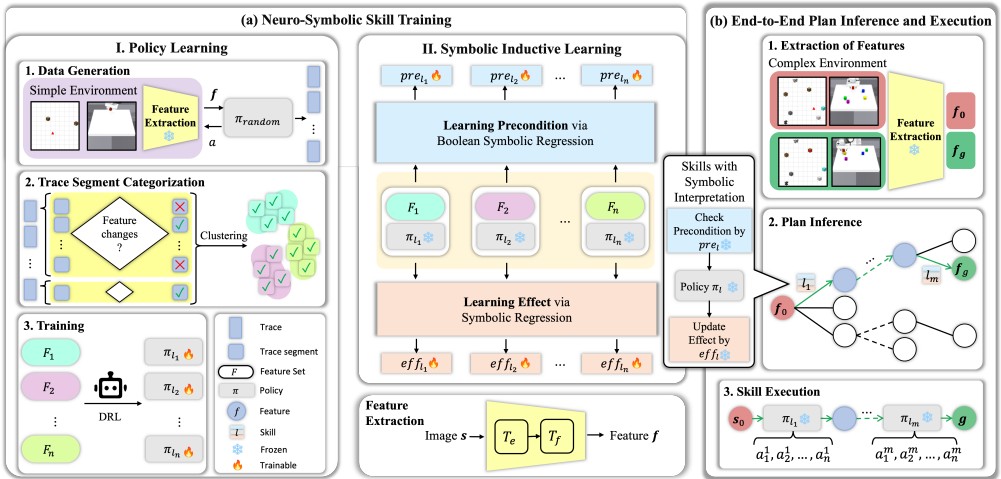

Figure 1: **Overview of the ReSIP Framework. (a) Neuro-Symbolic Skill Training:** First, policies for skills are learned by collecting traces from random policies, dividing these traces into segments based on feature changes, and clustering these segments into distinct sets to train corresponding policies. Next, symbolic regression is applied to these set-policy pairs to derive symbolic interpretations of each skill, explicitly learning their preconditions and effects. **(b) End-to-End Plan Inference and Execution:** Given initial and goal state images, the framework extracts corresponding features, then leverages MCTS to infer a valid plan of skills by satisfying symbolic preconditions and effects. Eventually, this symbolic plan guides action execution, enabling successful goal completion.

object-centric representation, We can decompose the representation $e$ into several sub-representation $\{e^i\}_{i=1}^m$ based on the $i$-th object. The sub-representation is used when the skill tries to achieve a subgoal. Details of DLP pretraining are in Appendix B.1.

**Feature Representation.** Inspired by the entity-centric architecture (Haramati et al., 2024), we develop an aggregation transformer as the second layer $T_f$ to aggregate entities into features for planning. The aggregation transformer comprises self-attention (SA) and cross-attention (CA) as its core components. SA is intended to extract *important attributes from the observation* more effectively, while CA is designed to capture the *temporal difference* between current state entities. The set of entities $\{e_n\}_{n=1}^N$ are processed by a sequence of Transformer (Vaswani, 2017) blocks: SA $\rightarrow$ CA $\rightarrow$ SA, followed by a MLP (Murtagh, 1991). A detailed architecture is depicted in Figure 6. The aggregation transformer is trained to minimize the mean square loss:

$$\mathcal{L}_{AT}(\hat{\boldsymbol{f}}) = \frac{1}{N} \sum_{i=1}^N \left(\boldsymbol{f}_i - \hat{\boldsymbol{f}}_i\right)^2, \tag{1}$$

where $N$ is the total number of training data. $\hat{\boldsymbol{f}}_i$ is the ground truth feature value.

When training the feature extraction module, we randomly initialize various parameters in the environment, such as the number and shapes of objects. Once these parameters are set, the system generates an image corresponding to the configured environment. We treat these randomly initialized parameters as ground truth $\hat{\boldsymbol{f}}_i$ and use Equation 1 to train the feature extractor. We have the flexibility to define a large number of features that we expect to be useful when describing the task.

## 3.2 SKILL LEARNING

Given a composite task, we aim to tackle it with a combination of simple skills. Our model is capable to learn the atomic skills $l = \langle args, \pi_l, pre_l(\boldsymbol{f}, o), eff_l(\boldsymbol{f}, o)\rangle$ from scratch, which relies on using the collected trace, composed of the original state, as training data. The key idea is that we first collect play data from interaction with the simple environment, and then we segment the data, which is the trace of the agent's movements, into trace segments and categorize these trace segments according to the feature change. Finally, we train the agent to learn the skill policy for each set of trace segments.

**Data Generation.** We first collect a significant amount of *play data* following previous work (Lynch et al., 2020; Rosete-Beas et al., 2023). Instead of struggling with the complex environments where

our agent should work during evaluation, we collect these traces from variant and simple training environments, such as environments with a single object in IsaacGym. Details are in Appendix B.2.

**Trace Segment Categorization.** After data generation, we divide and categorize the trace segments into different sets by their feature, getting the offline dataset based on different feature changes. Firstly, we divide the trace into trace segments of length $h$. Then, we retain only those trace segments where feature changes are observed. Finally, we employ a K-means clustering algorithm (Ahmed et al., 2020) to gather trace segments with similar feature changes, with the objective function:

$$\arg\min_{\mathcal{T}} \sum_{i=1}^{\mathcal{K}} \frac{1}{|\mathcal{T}_i|} \sum_{\tau_1, \tau_2 \in \mathcal{T}_i} \|\tau_1 - \tau_2\|_2 \tag{2}$$

where $\tau = [T_f(e_1), ..., T_f(e_h)]$ is the concatenation of trace segments in feature representation, $\mathcal{T}_i \subseteq \mathcal{T}$ is each classified cluster, and $\mathcal{K}$ is the total number of clusters.

**Training.** It is worth noting that the training algorithm for goal-conditioned policy $\pi_l$ is agnostic of the planning framework. As we categorize offline datasets into $\mathcal{K}$ clusters of trace segments, we adopt the goal-conditioned behavior cloning (GCBC) algorithm (Lynch et al., 2020) to learn a skill policy for each cluster. The network of the policy is also composed of transformer blocks. The outline of the policy network is a composed structure of SA and CA, which can model the relationship between the current state and the goal. We apply the GCBC loss to train each policy $\pi_l$ that can achieve the best performance. With the trace segments $\{s_1, ..., s_T\}$ and the object $o$, the loss is as following:

$$\mathcal{L}_{GCBC} = -\frac{1}{T} \sum_{t=1}^{T} \log(\pi_l(\boldsymbol{a}_t|\boldsymbol{s}_t, \boldsymbol{s}_T, o)), \tag{3}$$

During skill learning, some trivial skills might be learned. However, these skills do not affect the selection of the planning algorithm. Details can be referred to in Section 3.4.

### 3.3 SYMBOLIC INDUCTIVE LEARNING

To construct a plan using atomic skills, we derive symbolic interpretation for each skill, enabling compositional reasoning and task decomposition. These interpretations define the preconditions and effects of skills through mathematical formulas: preconditions are expressed as constraints, and effects are expressed as transformations, both using the operation set $\{+, -, \times, \div, >\}$ to form polynomial expressions.

**Symbolic Regression.** Given a skill $l$, the induction module proceeds to search for the effect $eff_l(\boldsymbol{f}, o)$ and precondition $pre_l(\boldsymbol{f}, o)$ for this skill policy. Since precondition and effect are functions of two classes of variables, $\boldsymbol{f}$ and $o$, where $\boldsymbol{f}$ and $o$ may be discrete, using symbolic regression becomes the most ideal method to find preconditions and effects. As Definition 2.2 states, the effect might change as the input state changes. Then we have $\boldsymbol{f}'_{final} = eff_l(\boldsymbol{f}_{init}, o)$, which can be formulated as a symbolic regression problem.

For symbolic regression, we use the PySR (Cranmer, 2023), which is a multi-population evolutionary algorithm. PySR is capable of performing feature selection, identifying the most significant element within the feature vector $\boldsymbol{f}$. Moreover, it also supports customizing the operator and the loss function.

**Precondition Rule.** Since the features in the environment might be complicated, determining whether a skill can be applied in the current stage is challenging. Here the Symbolic Regression module PySR outputs a boolean result using operators $\{>, =\}$. We train the symbolic regression module with a batch of collected features. We design a loss function for training:

$$\mathcal{L}_{SR}^{pre} = \sum_{i=1}^{N} \|b_{pred}^i - b_{target}^i\|_2, \tag{4}$$

where $b_{pred}^i$ is the predicted value and $b_{target}^i$ is the ground truth value.

**Effect.** For the effect of a skill $eff_l(\boldsymbol{f}, o)$, we mainly use constants and the binary operators $\{+, -, \times, \div\}$ to form the effect function. We design an element-wise loss function:

$$\mathcal{L}_{SR}^{eff} = \|\boldsymbol{f}_{pred} - \boldsymbol{f}_{target}\|_2 + complexity, \tag{5}$$

where $\boldsymbol{f}_{pred}$ is the prediction result and $\boldsymbol{f}_{target}$ is the ground truth. We employ the normalization term $complexity$ (Cranmer, 2023) to prioritize the effect function using a simple mathematical format.

### 3.4 END-TO-END PLAN INFERENCE AND EXECUTION

In this section, we introduce the overall process of end-to-end pixel-based planning given an initial image and a goal image. Figure 1(b) shows the complete process.

**Extraction of Features.** Given images of the initial state $\boldsymbol{s}_0$ and goal state $\boldsymbol{g}$, the feature extraction module introduced in Section 3.1 converts them into features $\boldsymbol{f}_0$ and $\boldsymbol{f}_g$.

**Plan Inference.** This part focuses on adopting Monte Carlo Tree Search (MCTS) (Kocsis & Szepesvári, 2006) to generate a ground skill plan $\Pi$ that fits the input feature $\boldsymbol{f}_0$ and the goal feature $\boldsymbol{f}_g$. We apply the Upper Confidence Bound applied to Trees (UCT) (Kocsis & Szepesvári, 2006):

$$\text{UCT}(l, o) = \frac{N(\boldsymbol{f}, l, o)_{succ}}{N(\boldsymbol{f}, l, o)} + C\sqrt{\frac{\ln N(\boldsymbol{f})}{N(\boldsymbol{f}, l, o)}}, \tag{6}$$

where $N(\boldsymbol{f}, l, o)_{succ}$ represents the success times of selecting ground skill $l(o)$ under feature $\boldsymbol{f}$, $C$ is a hyperparameter to balance the exploration and exploitation, $N(\boldsymbol{f})$ is the number of times feature $f$ has been visited in previous iterations, and $N(\boldsymbol{f}, l)$ is the number of times skill $l$ has been sampled in in feature $f$.

$$(l_i^*, o_j^*) = \arg\max_{l_i, o_j} UCT(l_i, o_j) * pre_{l_i}(\boldsymbol{f}, o_j). \tag{7}$$

As shown in Eq 7, we select the next skill for the node with the max $\text{UCT}(l_i, o_j)$, meanwhile ensuring the skill's precondition is true. Otherwise, we meet a terminal node. Then, we expand this skill if there are untried skills. Finally, we simulate some steps and update the $f_p$ of each node according to the reward, the number of visits, and UCT. After many rounds, we obtain the ground skill plan $\Pi$.

**Skill Execution.** For each ground skill $l(o)$, we have an input image $\boldsymbol{s}^i$, which represents the current state, and goal image $\boldsymbol{g}^i$. And we set the policy time horizon as $t$. A skill can execute for consecutive $t$ timesteps before switching to the next one. During the execution, the policy output an action $\pi_l(\boldsymbol{a}|\boldsymbol{s}^i, \boldsymbol{g}^i, o)$. Thus, we find an approach to accomplish the whole task.

## 4 EXPERIMENTS

To evaluate the performance of ReSIP, we select two different types of environments. One is the Minecraft environment, which verifies a series of long-horizon compositional tasks. The other is Isaac-Gym, a robotic arm simulation environment employed to assess the performance of compositional generalization tasks. ReSIP trains atomic skills from single-object demonstrations and composes them to solve tasks in complex multi-object environments, demonstrating strong generalization capabilities in both long-horizon sequential tasks and object manipulation.

**Environments.** Minecraft is an $n \times n$ grid world environment. It is inspired by the computer game Minecraft and is similar to the environment in previous works (Brooks et al., 2021; Hasan-beig et al., 2021; Kokel et al., 2021; Liu et al., 2024). An agent can move along four directions $\{\text{up}, \text{down}, \text{left}, \text{right}\}$ and interact with objects with learned skills. Different from the previous environment, our inputs are image maps with different objects in the map. We evaluate ReSIP on a suite of tasks with increasing complexity. `Make-Stick` serves as a basic task, requiring the agent to produce a single stick. `Make-Mass-Sticks` extends this by demanding repeated execution of the same skill to produce multiple sticks, testing skill reuse. `Pickup-Iron` introduces tool dependencies, requiring the agent to craft several tools before completing the goal. `Multiple-Goals` involves collecting four items in a predefined order, emphasizing multi-step sequencing. Finally, `Make-Enhance-Table` presents the most challenging long-horizon scenario, where the agent must orchestrate many interdependent skills to construct an enhanced table.

IsaacGym (Makoviychuk et al., 2021) is a simulated tabletop robotic object manipulation environment. The environment includes a robotic arm set in front of a table with various cubes and buttons in different colors. The agent observes the system's state through visual input and performs actions in the form of deltas in the end effector coordinates $a = (\Delta x, \Delta y, \Delta z, \Delta g)$, where $\Delta g$ indicates whether

the gripper is open or closed. At the beginning of each episode, both the current cube positions and the goal positions are randomly initialized on the table. The tasks are as follows: `Push` requires the agent to push cubes with **randomized numbers and color** to the goal location. `Push-Grab-Lift` needs the agent to manipulate cubes of randomized numbers and color to their goal positions by **pushing and lifting** operation. In `Ordered-Press`, the agent should press different buttons **in a particular order**.

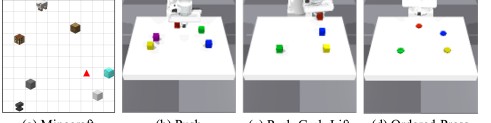 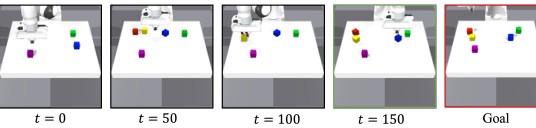

(a) Minecraft    (b) Push    (c) Push-Grab-Lift    (d) Ordered-Press      $t = 0$    $t = 50$    $t = 100$    $t = 150$    Goal

Figure 2: The environments used for experiments in this work.

Figure 3: A sample trace of an agent in object manipulation `Push`.

**Baselines.** We compare our framework with various DRL algorithms with pixel-based decision transformers (ECRL, SMORL) learning from *rewards*, imitation learning methods (GAIL) learning from *demonstration*, and methods that combine planning and DRL (Deepsynth, DiRL). **SMORL** (Zadaianchuk et al., 2020) adopts object-centric representations with goal-conditioned attention policies to discover and learn useful skills. **ECRL** (Haramati et al., 2024) uses object-centric representations with the entity-interaction transformer to discover and learn useful skills. **GAIL** (Ho & Ermon, 2016) mimics expert behaviors via learning a generative adversarial network whose generator is a policy. **DeepSynth** (Hasanbeig et al., 2021) uses an automaton to find the substructure of tasks and execute the subtasks using the low-level controller.**DiRL** (Jothimurugan et al., 2021) uses a predefined logical specification to decompose tasks into subtasks then solve them by DRL controllers.

### 4.1 LONG-HORIZON SEQUENTIAL TASK

We evaluate the different methods in the Minecraft Environment to test the performance in some long-horizon tasks. Results are presented in Table 1. For long-horizon planning tasks, **ECRL, SMORL** achieve a low success rate because they cannot handle temporal logic tasks. **Deepsynth** uses an automaton-based high-level structure for task decomposition, so it has a relatively high success rate in simple tasks. However, as the tasks become complex, their performance drops sharply since the search space for the automaton is too big for the algorithm to cover. Figure 4 demonstrate an example of end-to-end pixel-based planning of `Make-Stick`.

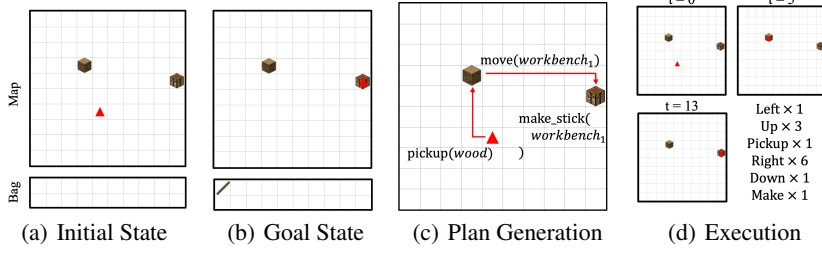

(a) Initial State      (b) Goal State      (c) Plan Generation      (d) Execution

Figure 4: End-to-end pixel-based planning of `Make-Stick`.

### 4.2 OBJECT MANIPULATION

We evaluate different methods in the IsaacGym and present results in Table 2. Here, we mainly report the overall success rate. For `Push`, we observe that most of the structured baselines **ECRL, SMORL** can achieve a high success rate. In contrast, conventional behavior cloning **GAIL** performs poorly as the number of cubes increases due to its poor compositional generalizability. **Deepsynth** and **DiRL** use the idea of task decomposition, however, the decompositional logic is simple and relies on expert knowledge. Thus, they also perform poorly as the number of cubes increases. `Push-Grab-Lift` and `Ordered-Press` have some logical dependency on their subtasks. The performance of **ECRL,**

| Task | Make-Stick | Make-Mass-Sticks | Pickup-Iron | Multiple-Goals | Make-Enhance-Table |
|---|---|---|---|---|---|
| SMORL | $51.1_{\pm0.5}$ | $37.4_{\pm1.0}$ | $33.2_{\pm1.2}$ | $28.1_{\pm0.9}$ | $11.2_{\pm1.1}$ |
| ECRL | $46.2_{\pm0.7}$ | $35.8_{\pm1.2}$ | $37.1_{\pm1.2}$ | $29.7_{\pm1.0}$ | $18.7_{\pm1.2}$ |
| GAIL | $51.3_{\pm0.5}$ | $46.9_{\pm0.9}$ | $41.5_{\pm1.1}$ | $30.2_{\pm1.1}$ | $16.4_{\pm1.2}$ |
| DiRL | $86.7_{\pm0.5}$ | $86.3_{\pm0.5}$ | $79.4_{\pm0.7}$ | $71.9_{\pm0.7}$ | $58.1_{\pm0.9}$ |
| DeepSynth | $90.1_{\pm0.3}$ | $85.3_{\pm0.5}$ | $82.1_{\pm0.5}$ | $69.3_{\pm0.8}$ | $55.7_{\pm1.0}$ |
| **Ours** | $\mathbf{95.8_{\pm0.4}}$ | $\mathbf{93.8_{\pm0.4}}$ | $\mathbf{91.7_{\pm0.5}}$ | $\mathbf{88.7_{\pm0.8}}$ | $\mathbf{75.0_{\pm0.8}}$ |

Table 1: **Success rates for long-horizon sequential tasks in Minecraft.**

| | Cubes | 1 | 2 | 3 | 4 | 5 |
|---|---|---|---|---|---|---|
| | SMORL | $99.0_{\pm0.0}$ | $83.8_{\pm0.4}$ | $50.9_{\pm1.0}$ | $43.8_{\pm1.2}$ | $30.2_{\pm1.2}$ |
| | ECRL | $97.3_{\pm0.5}$ | $96.3_{\pm0.5}$ | $83.8_{\pm0.4}$ | $72.3_{\pm0.6}$ | $57.0_{\pm1.0}$ |
| Push | GAIL | $95.5_{\pm0.5}$ | $75.0_{\pm0.4}$ | $47.8_{\pm1.2}$ | $43.8_{\pm1.1}$ | $39.6_{\pm1.1}$ |
| | DiRL | $93.5_{\pm0.5}$ | $87.3_{\pm0.6}$ | $74.1_{\pm0.5}$ | $61.2_{\pm0.9}$ | $50.5_{\pm1.2}$ |
| | DeepSynth | $91.9_{\pm0.3}$ | $86.1_{\pm0.3}$ | $80.3_{\pm0.5}$ | $63.5_{\pm0.5}$ | $49.3_{\pm0.6}$ |
| | **Ours** | $\mathbf{100_{\pm0.0}}$ | $\mathbf{100_{\pm0.0}}$ | $\mathbf{87.5_{\pm0.5}}$ | $\mathbf{75.0_{\pm0.4}}$ | $\mathbf{68.8_{\pm0.7}}$ |
| | SMORL | $25.0_{\pm0.6}$ | $0.0_{\pm0.0}$ | $0.0_{\pm0.0}$ | $0.0_{\pm0.0}$ | $0.0_{\pm0.0}$ |
| Push- | ECRL | $25.0_{\pm0.8}$ | $0.0_{\pm0.0}$ | $0.0_{\pm0.0}$ | $0.0_{\pm0.0}$ | $0.0_{\pm0.0}$ |
| Grab- | GAIL | $48.8_{\pm0.9}$ | $34.8_{\pm1.2}$ | $10.1_{\pm0.9}$ | $3.1_{\pm0.8}$ | $1.0_{\pm0.6}$ |
| Lift | DiRL | $52.1_{\pm1.0}$ | $35.5_{\pm1.1}$ | $23.5_{\pm1.1}$ | $6.5_{\pm1.1}$ | $2.5_{\pm0.9}$ |
| | DeepSynth | $50.2_{\pm0.7}$ | $33.0_{\pm1.0}$ | $18.7_{\pm1.3}$ | $8.3_{\pm0.9}$ | $0.0_{\pm0.0}$ |
| | **Ours** | $\mathbf{62.5_{\pm0.7}}$ | $\mathbf{50.0_{\pm0.8}}$ | $\mathbf{50.0_{\pm0.9}}$ | $\mathbf{15.6_{\pm0.5}}$ | $\mathbf{12.5_{\pm0.7}}$ |
| | SMORL | $98.0_{\pm0.4}$ | $68.6_{\pm0.8}$ | $51.3_{\pm1.1}$ | $37.2_{\pm1.1}$ | $15.8_{\pm1.2}$ |
| Ordered- | ECRL | $\mathbf{100.0_{\pm0.0}}$ | $62.5_{\pm0.8}$ | $42.7_{\pm1.1}$ | $35.4_{\pm1.0}$ | $27.7_{\pm1.5}$ |
| Press | GAIL | $97.1_{\pm0.3}$ | $86.2_{\pm0.4}$ | $69.7_{\pm0.8}$ | $53.5_{\pm1.1}$ | $32.8_{\pm1.2}$ |
| | DiRL | $98.2_{\pm0.4}$ | $93.1_{\pm0.5}$ | $84.1_{\pm0.5}$ | $70.3_{\pm0.5}$ | $66.2_{\pm1.0}$ |
| | DeepSynth | $92.5_{\pm0.5}$ | $90.8_{\pm0.6}$ | $80.3_{\pm0.5}$ | $65.5_{\pm0.7}$ | $53.5_{\pm1.0}$ |
| | **Ours** | $99.0_{\pm0.4}$ | $\mathbf{93.8_{\pm0.4}}$ | $\mathbf{87.5_{\pm0.6}}$ | $\mathbf{81.3_{\pm0.5}}$ | $\mathbf{81.3_{\pm0.6}}$ |

Table 2: **Success rates for object manipulation tasks in IsaacGym.**

**SMORL** is much poorer than our model because these two models have no awareness of the temporal attributes of sub-tasks, which shows the superiority of our skills with symbolic interpretation. Figure 3 shows the sample trace of `Push`. Other detailed results are in Appendix D.3.

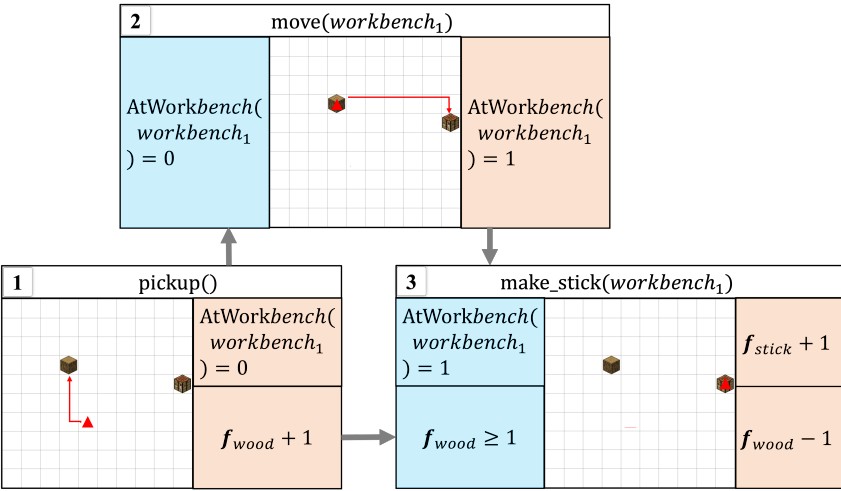

Figure 5: Skill relation based on symbolic interpretation in `Make-Stick`. Expressions in the blue box represent the *precondition*, and orange denotes the *effect*.

### 4.3 Symbolic Interpretation

Symbolic interpretation is an essential feature of our skills, which enables the searching algorithm to find a feasible plan for a complex task. We have shown the symbolic interpretation of IsaacGym in Table 5 and Minecraft in Table 4, where the preconditions are the boolean formula and the effects are in the format of a function. Figure 5 demonstrates the symbolic interpretation in `Make-Stick`.

Taking `Make-Stick` as an example, the agent first compares the initial images and goal images demonstrating its task and is aware that it should make a stick at the workbench as shown in Figure 5. The preconditions of the last action $\mathrm{make(stick)}$ are $\mathrm{wood} \geq 1$ and $\mathrm{at\_workbench} = 1$, which means the agent should move to the workbench with a wood. The effect of $\mathrm{move(workbench)}$ is $\mathrm{at\_workbench} = 1$, thus we have $\mathrm{move(workbench)} \prec \mathrm{make(stick)}$. Similarly, the effect for $\mathrm{pickup(wood)}$ is $\mathrm{wood} + 1$, thus we have $\mathrm{pickup(wood)} \prec \mathrm{make(stick)}$. We can get this relation from the dependency graph in Figure 8. In plan generation, the agent induces a symbolic plan by MCTS, forming a sequence: $\mathrm{pickup(wood)} \rightarrow \mathrm{move(workbench)} \rightarrow \mathrm{make(stick)}$, which satisfies the aforementioned partial order relation and can accomplish the task, thus forming our final plan.

## 5 Related Work

**Object-Centric RL.** Many recent works employed the structured representation in model-free and model-based RL (Colas et al., 2019; Zadaianchuk et al., 2022; Mambelli et al., 2022; Zhao et al., 2022; Zhou et al., 2022; Ferraro et al., 2023; Feng & Magliacane, 2024). Among them, methods such as SMORL (Zadaianchuk et al., 2020) and ECRL (Haramati et al., 2024) leverage object-centric representations (Jiang et al., 2019; Francesco et al., 2020; Daniel & Tamar, 2022) in combination with goal-conditioned attention policies to discover and learn useful skills from raw image data. However, they cannot segment learned skills into atomic units and reform them to solve novel and complex tasks. In this work, we integrate RL into symbolic inductive planning, thus giving our method the ability to learn atomic skills and understand how to compose them in long-horizon tasks.

**Neuro-Symbolic RL.** Several works explore utilizing symbolic methods in DRL to deal with robotic tasks (Belta et al., 2007; Blaes et al., 2019; Illanes et al., 2020; Kokel et al., 2021; Sehgal et al., 2023; Silver et al., 2023; Acharya et al., 2024), including planning domain definition language (Mao et al., 2023), automata (Hasanbeig et al., 2021), Spectrl (Jothimurugan et al., 2021; Žikelić et al., 2024). These methods require either predefined symbolic structures or predefined skills, limiting their compositional generalizability to complex object manipulation. In contrast, our framework can automatically discover skills from multiple single-object environments and utilize symbolic interpretation to reform these skills for novel and complex tasks in multi-object environments.

## 6 Conclusion

We present a model combining a planning framework with DRL to solve pixel-based control challenges. Our model can autonomously acquire atomic skills through interaction with the environment, minimizing the need for expert knowledge. Moreover, by providing symbolic interpretations for skills, we can form a ground skill plan for long-horizon tasks through search algorithms such as MCTS. Additionally, our model leverages composable skills and a transformer-based action policy, which provides compositional generalizability to tasks that share similar features. Our model has shown great performance on long-horizon, pixel-based control problems based on this superiority.

**Limitation.** Our model requires a pre-trained image segmentation model. While basic image segmentation models have achieved promising results in our experiments, more complex tasks may require further advancements and refinements in image semantic segmentation techniques. Additionally, the approach using discrete features as an interface may induce some inaccuracy and inflexibility. Some states with slight differences may share the same feature representation.

**Future Work.** For future work, one interesting direction is to explore more advanced ways for automatic skills generation. Currently, the skill generation relies on classifying the collected traces. We can further improve it with reward-based or entropy-based methods in the future. Another possible direction is to employ generative models, such as diffusion models, to replace the current image segmentation approach to generate sub-goal images.

ETHICS STATEMENT

This work adheres to the ICLR Code of Ethics. In this study, no human subjects or animal experimentation were involved. All environments used, including the IsaacGym and Minecraft, were sourced in compliance with relevant usage guidelines, ensuring no violation of privacy. We have taken care to avoid any biases or discriminatory outcomes in our research process. No personally identifiable information was used, and no experiments were conducted that could raise privacy or security concerns. We are committed to maintaining transparency and integrity throughout the research process.

REPRODUCIBILITY STATEMENT

We have made every effort to ensure that the results presented in this paper are reproducible. All code and datasets have been made publicly available in an anonymous repository to facilitate replication and verification. The experimental setup, including training steps, model configurations, and hardware details, is described in detail in the paper. We have also provided a full description of our ReSIP framework to assist others in reproducing our experiments. We believe these measures will enable other researchers to reproduce our work and further advance the field.

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

## A    ALGORITHM

We outline the algorithm of our end-to-end pixel-based framework below. Lines 2 to 15 detail the preprocessing of traces, involving categorizing them into distinct groups for subsequent training. Lines 16 to 24 describe the process of skill formation through symbolic interpretation of the traces. Lines 25 to the end encompass the planning and execution of different tasks.

**Algorithm 1** The whole training and evaluation of the framework.

**Input:** The total trace collecting step $N$. The evaluation task $\mathbb{T}_{eval}$.
Randomly initialize some simple environment $e_i$
**for** $j \leftarrow 0$ **to** N **do**
   Interact with simple environment with random policy $\pi_{random}$
   Collect the trace $\tau_{ori}$
**end for**
**for** $\tau_{ori} \in$ T **do**
   $p \leftarrow 0, q \leftarrow 0$
   **while** $T_f(T_e(\boldsymbol{s}_p)) = T_f(T_e(\boldsymbol{s}_q))$ **do**
      $q$++
   **end while**
   Segment the trace $\tau_{ori}[p : q]$
   Insert the trace to a trace set $\mathcal{S}_{trace}$
   p = q
**end for**
Classify the trace using cluster algorithm
**for** trace cluster $\mathcal{S}_i \in \mathcal{S}_{trace}$ **do**
   Randomly initialize policy $\pi_i$
   Training policy $\pi_i$ with GCBC algorithm
**end for**
**for** $i \leftarrow 0$ **to** $|\mathcal{S}_{trace}|$ **do**
   Find the effect of $\pi_i$ through PySR
   Find the precondition of $\pi_i$ through neural guidance algorithm
   Form a skill with symbolic interpretation $l(\boldsymbol{s}) = \langle \boldsymbol{s}, o, \pi_l, pre_l(\boldsymbol{s}, f), \mathit{eff}_l(\boldsymbol{s}, f) \rangle$
**end for**
Get the initial state $\boldsymbol{s}_{init}$ and goal state $\boldsymbol{g}$ of $\mathbb{T}_{eval}$
Get the initial feature $f_{init}$ and goal feature $f_g$
Using MCTS to find a path $l_1 \rightarrow l_2 \rightarrow \cdots \rightarrow l_n$ from $f_{init}$ to $f_g$
**for** $j \leftarrow 0$ **to** n **do**
   **for** $t \leftarrow 0$ **to** volley **do**
      Get the action $\boldsymbol{a} = \pi_i(\boldsymbol{s}_{j \times volley + t})$
      Interact with the environment $\boldsymbol{s}_{j \times volley + t + 1} = \mathbb{T}_{eval}(\boldsymbol{a})$
   **end for**
**end for**

# B IMPLEMENTATION DETAILS

## B.1 PRE-TRAINED MODELS

**Object-Centric Representation.** Our OCR algorithm is based on the DLP algorithm. DLP (Daniel & Tamar, 2022) is an unsupervised object-centric model for images based on variational autoencoder (VAE) (Kingma, 2013). It provides the latent representation for all the particles.

The foreground representation $e = [e_c, e_s, e_d, e_t, e_f] \in \mathbb{R}^{11}$ is a disentangled latent variable including the following learned attributes: spatial coordinate $e_c \in \mathbb{R}^3$, scale $e_s \in \mathbb{R}^2$, depth $e_d \in \mathbb{R}$, transparency $e_t \in \mathbb{R}$, and visual features $e_f \in \mathbb{R}^4$. Here we set the number of entities as 24.

**Aggregation Function.** We design an aggregation transformer inspired by entity-centric architecture (Haramati et al., 2024), which processes the OCR entities into features of the environment. An architecture outline is presented in Figure 6. The aggregation transformer comprises self-attention (SA) and cross-attention (CA) as its core components. The self-attention tries to grab the relation of entities in a single object. It transforms the input vector, grouping all the entities of a single object. At the CA layer, the transformer network tries to figure out the relation between different objects. After passing the SA and CA network, we let the model pass another (SA) network again. This network considers the result from the previous steps and forms these two steps together, placing self-attention calculation on the overall computing result. Finally, we get the output result, which is the feature.

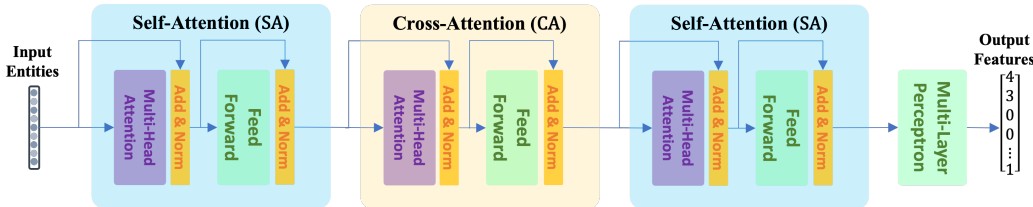

Figure 6: The architecture of the aggregation transformer.

## B.2 SKILL LEARNING

It is worth mentioning that our framework is agnostic of the skill policy. We have tried several RL algorithms and finally chose GCBC (Lynch et al., 2020) since it has the best performance. We use GCBC to train the policy $\pi_l$ for the skill. This method extracts goal-conditioned policies using self-supervision on top of raw, unlabeled data.

As mentioned in the section 3.2, we collect traces from interaction with the simple environment. Taking the IsaacGym environment as an example, we set the object number to be one, the object type to be a cube, and the object color to be red. The agent can operate its gripper to interact with the only object that appeared on the table. Thus, it can collect a tremendous amount of data.

Figure 7 shows the learning curves of the skills, the y-axis is the success rate during the training process. The names and functions of skills are not specified in advance. We name the skill according to its effect.

## B.3 EFFECTS OF SKILL

We use PySR (Cranmer, 2023) for the implementation of the symbolic regression part of the skills, generating the mathematical form of the effect. In PySR, we can use specific parameters to control the generation of the formula. The parameter settings for the regressor are in Table 3.

Here, we assume that all the effects of skills on features can be characterized by some polynomial expression. Then, we use the binary operation and the constants to form such a relation. We treat the initial feature and final feature of a sequence as the input and output of a function. Then we fit the relationship between tuples of input and output.

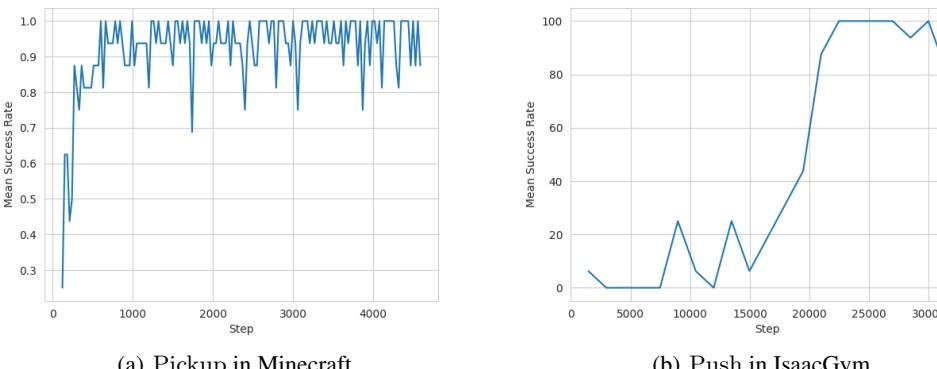

| (a) Pickup in Minecraft | (b) Push in IsaacGym |
|---|---|

Figure 7: The curves of mean success rate during the skill learning process under 96 random goals in IsaacGym and Minecraft environments. Notice that the names and functions of skills are not specified in advance.

| Parameters | Value |
|---|---|
| Number of Iterations | 40 |
| Complexity | 5 |
| Binary Operators | $\{+, -, \times, \div\}$ |
| Unary Operator | $\{>\}$ |
| fractionReplaced | 0.1 |
| shouldOptimizeConstants | True |
| maxsize | 20 |
| procs | 4 |

Table 3: The Parameter Setting of PySR

## C  ENVIRONMENT SETTING DETAILS

### C.1  BASELINE REIMPLEMENTATION

**SMORL.**  We reimplemented SMORL (Zadaianchuk et al., 2020), substituting its original visual model SCALOR (Jiang et al., 2019) with DLP. Additionally, the low-level controller within the SMORL framework was replaced with the same controller used in our proposed method.

**ECRL.**  The original version of ECRL was used directly in our experiments.

**GAIL.**  GAIL (Ho & Ermon, 2016) was reimplemented with several modifications. We integrated DLP to process image input and replaced the actor with the same controller used in our framework. Furthermore, the critic and discriminator networks within GAIL were updated to employ a transformer architecture.

**DeepSynth.**  We reimplement DeepSynth (Hasanbeig et al., 2021), with the original image segmentation algorithm replaced by DLP. We directly implement the automaton synthesis algorithm based on the DLP result. For the low-level controller in DeepSynth, we also use the same controller as our framework to substitute the controller in DeepSynth to ensure a fair comparison.

**DiRL.** DiRL (Jothimurugan et al., 2021) was reimplemented as a baseline model, incorporating domain-specific knowledge. Rules such as "pick after push" and "pick up wood before going to the craft table" were established to provide high-level guidance for the low-level policy. The policy within DiRL was also replaced with the same controller used in our framework for a more credible comparison.

## C.2 MINECRAFT

We design the features to extract as follows:

- at_wood: A boolean variable representing whether the agent's position is at wood.
- at_stone: A boolean variable representing whether the agent's position is at stone.
- at_iron: A boolean variable representing whether the agent's position is at iron.
- at_gem: A boolean variable representing whether the agent's position is at gem.
- at_sheep: A boolean variable representing whether the agent's position is at sheep block.
- at_workbench: A boolean variable representing whether the agent's position is at the workbench.
- at_toolshed: A boolean variable representing whether the agent's position is at the toolshed.
- wood: The number of wood in the agent's bag.
- stone: The number of stones in the agent's bag.
- iron: The number of iron in the agent's bag.
- gem: The number of gems in the agent's bag.
- stick: The number of sticks in the agent's bag.
- stone_pickaxe: The number of stone pickaxes in the agent's bag.
- iron_pickaxe: The number of iron pickaxes in the agent's bag.
- scissors: The number of scissors in the agent's bag.
- paper: The number of paper in the agent's bag.
- wool: The number of wool in the agent's bag.
- enhance_table: The number of enhanced tables in the agent's bag.
- bed: The number of beds in the agent's bag.
- jukebox: The number of jukeboxes in the agent's bag.

As we have form skills with symbolic interpretation, we can use a graph to describe the dependency relation between different skills. A detailed dependency graph of all the skills in Minecraft is shown in Figure 8.

## C.3 ISAACGYM

We design the features of IsaacGym to extract as follows:

- num_objects: The number of objects on the table captured by cameras that provide front view and side view.
- xy_goal: The number of objects reaching their goals on the table.
- z_goal: The number of objects reaching their goals in the air lifted by the gripper.
- is_grab: A boolean variable representing whether the gripper grabs the object.
- color_1, . . . , color_5: A boolean variable representing whether a color exists. It can record at most five colors.
- next_color: It is an integer that stands for the next color that should be controlled. It guarantees the ordered operation of objects following the color sequence, which is red, green, blue, yellow, and purple in our case.

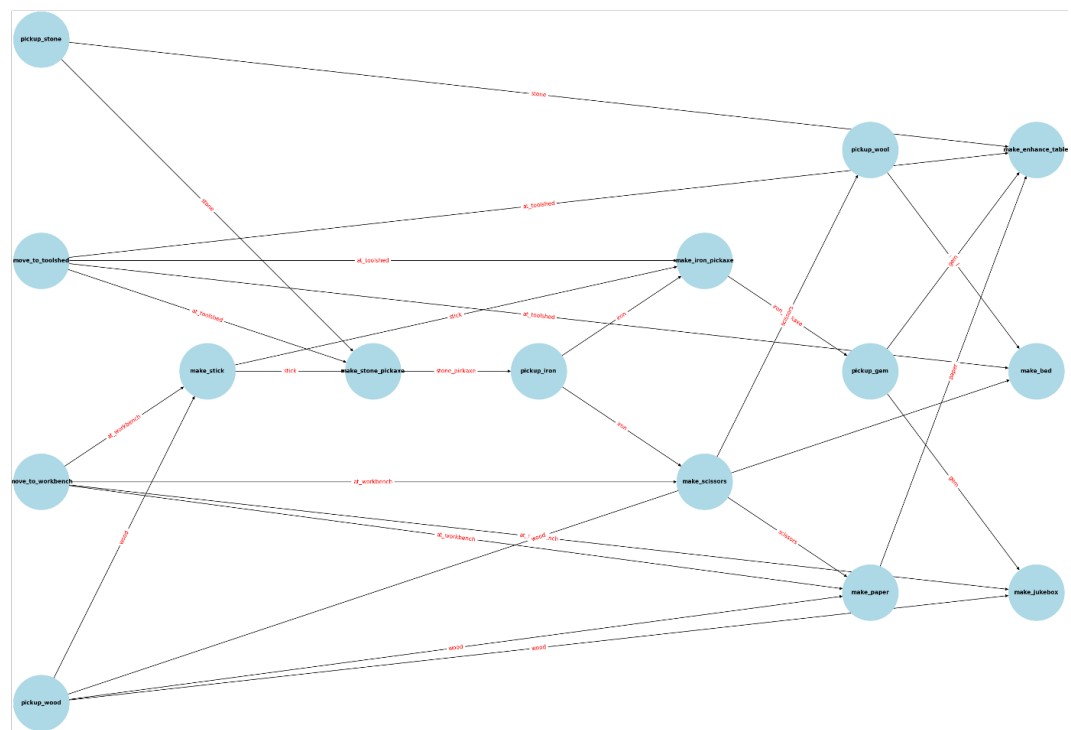

Figure 8: Dependency graph of Minecraft.

## C.4 EVALUATION METRICS

We mainly evaluate the performance of different methods based on the success rate and success fraction. Apart from these two results, we also record more detailed information in the experiment, including color success, color success fraction, action success, and action success fraction.

- **Success Rate.** The success rate describes the final success rate of the whole task.
- **Success Fraction.** The success fraction is the portion of accomplished subtasks, so this metric is usually higher than the success.
- **Color Success.** The color success gives out each color's success rate.
- **Color Success Fraction.** The color success fraction indicates the percentage of completed subtasks in each color.
- **Action Success.** The action success reflects the success rate of each action.
- **Action Success Fraction.** The action success fraction represents the percentage of accomplished subtasks in each action.

## D ADDITIONAL EXPERIMENTAL RESULTS

## D.1 ABLATION STUDY

In this section, we examine how certain key components of the model affect its performance.

### D.1.1 INFLUENCE OF THE NUMBER OF CLUSTERS

We investigate how the number of clusters $\mathcal{K}$ affects the result of the whole task. We take the object manipulation task as an example. In this task, the number of skills is 4. We set the $\mathcal{K}$ from 3 to 7 and investigate the success rate of the whole task. The tests are conducted on the environment Push-Grab-Lift which needs multiple skills. When the cluster number is set to 3, the task has an

| Skill | Object | Preconditions | Effects |
|-------|--------|---------------|---------|
| move | workbench | $\text{AtWorkbench}(\lambda_{\text{workbench}}) = 0$ | $\boldsymbol{f}_{at\_wood} = 0, \boldsymbol{f}_{at\_stone} = 0,$ $\boldsymbol{f}_{at\_iron} = 0, \boldsymbol{f}_{at\_gem} = 0,$ $\boldsymbol{f}_{at\_wool} = 0, \boldsymbol{f}_{at\_workbench} = 1,$ $\boldsymbol{f}_{at\_toolshed} = 0$ |
| | toolshed | $\text{AtToolshed}(\lambda_{\text{toolshed}}) = 0$ | $\boldsymbol{f}_{at\_wood} = 0, \boldsymbol{f}_{at\_stone} = 0,$ $\boldsymbol{f}_{at\_iron} = 0, \boldsymbol{f}_{at\_gem} = 0,$ $\boldsymbol{f}_{at\_wool} = 0, \boldsymbol{f}_{at\_workbench} = 0,$ $\boldsymbol{f}_{at\_toolshed} = 1$ |
| pickup | wood | $\text{AtMaterial}(\lambda_{\text{material}}) = 1$ | $\boldsymbol{f}_{wood} + 1$ |
| | stone | $\text{AtMaterial}(\lambda_{\text{material}}) = 1$ | $\boldsymbol{f}_{stone} + 1$ |
| | grass | $\text{AtMaterial}(\lambda_{\text{material}}) = 1$ | $\boldsymbol{f}_{grass} + 1$ |
| | bamboo | $\text{AtMaterial}(\lambda_{\text{material}}) = 1$ | $\boldsymbol{f}_{bamboo} + 1$ |
| | iron | $\text{AtMaterial}(\lambda_{\text{material}}) = 1 \wedge \boldsymbol{f}_{stone\_pickaxe} \geq 1$ | $\boldsymbol{f}_{iron} + 1$ |
| | gem | $\text{AtMaterial}(\lambda_{\text{material}}) = 1 \wedge \boldsymbol{f}_{iron\_pickaxe} \geq 1$ | $\boldsymbol{f}_{gem} + 1$ |
| | wool | $\text{AtMaterial}(\lambda_{\text{material}}) = 1 \wedge \boldsymbol{f}_{scissors} \geq 1$ | $\boldsymbol{f}_{wool} + 1$ |
| make_stick | workbench | $\text{AtWorkbench}(\lambda_{\text{workbench}}) = 0 \wedge \boldsymbol{f}_{wood} \geq 1$ | $\boldsymbol{f}_{stick} + 1, \boldsymbol{f}_{wood} - 1$ |
| make_grass_stack | workbench | $\text{AtWorkbench}(\lambda_{\text{workbench}}) = 0 \wedge \boldsymbol{f}_{grass} \geq 1$ | $\boldsymbol{f}_{grass\_stack} + 1, \boldsymbol{f}_{grass} - 1$ |
| make_bamboo_fence | workbench | $\text{AtWorkbench}(\lambda_{\text{workbench}}) = 0 \wedge \boldsymbol{f}_{bamboo} \geq 1$ | $\boldsymbol{f}_{bamboo\_fence} + 1, \boldsymbol{f}_{bamboo} - 1$ |
| make_stone_pickaxe | toolshed | $\text{AtToolshed}(\lambda_{\text{toolshed}}) = 0 \wedge$ $\boldsymbol{f}_{stick} \geq 2 \wedge \boldsymbol{f}_{stone} \geq 3$ | $\boldsymbol{f}_{stone\_pickaxe} + 1,$ $\boldsymbol{f}_{stick} - 2, \boldsymbol{f}_{stone} - 3$ |
| make_iron_pickaxe | toolshed | $\text{AtToolshed}(\lambda_{\text{toolshed}}) = 0 \wedge$ $\boldsymbol{f}_{stick} \geq 2 \wedge \boldsymbol{f}_{iron} \geq 3$ | $\boldsymbol{f}_{iron\_pickaxe} + 1,$ $\boldsymbol{f}_{stick} - 2, \boldsymbol{f}_{iron} - 3$ |
| make_scissors | workbench | $\text{AtWorkbench}(\lambda_{\text{workbench}}) = 0 \wedge \boldsymbol{f}_{iron} \geq 2$ | $\boldsymbol{f}_{scissors} + 1, \boldsymbol{f}_{iron} - 2$ |
| make_paper | workbench | $\text{AtWorkbench}(\lambda_{\text{workbench}}) = 0 \wedge$ $\boldsymbol{f}_{scissors} \geq 1 \wedge \boldsymbol{f}_{wood} \geq 1$ | $\boldsymbol{f}_{paper} + 1, \boldsymbol{f}_{wood} - 1$ |
| make_bed | toolshed | $\text{AtToolshed}(\lambda_{\text{toolshed}}) = 0 \wedge$ $\boldsymbol{f}_{wood} \geq 3 \wedge \boldsymbol{f}_{wool} \geq 3$ | $\boldsymbol{f}_{bed} + 1,$ $\boldsymbol{f}_{wood} - 3, \boldsymbol{f}_{wool} - 3$ |
| make_jukebox | workbench | $\text{AtWorkbench}(\lambda_{\text{workbench}}) = 0 \wedge$ $\boldsymbol{f}_{wood} \geq 3 \wedge \boldsymbol{f}_{gem} \geq 1$ | $\boldsymbol{f}_{jukebox} + 1, \boldsymbol{f}_{wood} - 3,$ $\boldsymbol{f}_{gem} - 1$ |
| make_enhance_table | workbench | $\text{AtWorkbench}(\lambda_{\text{workbench}}) = 0 \wedge \boldsymbol{f}_{stone} \geq 1 \wedge$ $\boldsymbol{f}_{paper} \geq 2 \wedge \boldsymbol{f}_{gem} \geq 1$ | $\boldsymbol{f}_{enhance\_table} + 1, \boldsymbol{f}_{stone} - 1,$ $\boldsymbol{f}_{paper} - 2 \wedge \boldsymbol{f}_{gem} - 1$ |

Table 4: Learned skills of the Minecraft environment.

| Skill | Object | Preconditions | Effects |
|-------|--------|---------------|---------|
| push | cube | | $\boldsymbol{f}_{xy\_goal} + 1$ |
| approach | cube | $\boldsymbol{f}_{is\_grab} = 0$ | $\boldsymbol{f}_{is\_grab} = 1$ |
| lift | cube | $\boldsymbol{f}_{is\_grab} = 1$ | $\boldsymbol{f}_{xy\_goal} + 1, \boldsymbol{f}_{z\_goal} + 1$ |
| press | button | $\boldsymbol{f}_{next\_color} < \boldsymbol{f}_{num\_objects}$ | $\boldsymbol{f}_{next\_color+1}$ |

Table 5: Learned skills of the IsaacGym environment.

extremely low success rate because the actual number of skills exceeds the cluster number, thus some skills are not learned. The detailed result is shown in Table 6.

### D.1.2 THE EFFECTIVENESS OF SYMBOLIC REGRESSION.

To show the effectiveness of our symbolic regression module with PySR, we replace the PySR module with neural network module. We added a rounding layer after the output layer of the neural network. The result of 3-cube tasks is in Table 7.

| Number of Cluster $\mathcal{K}$ | 3 | 4 | 5 | 6 | 7 |
|---|---|---|---|---|---|
| Push-Grab-Lift-1-Cube | 0.001 | 0.625 | 0.633 | 0.642 | 0.611 |
| Push-Grab-Lift-2-Cube | 0.002 | 0.500 | 0.512 | 0.493 | 0.499 |
| Push-Grab-Lift-3-Cube | 0.002 | 0.500 | 0.487 | 0.493 | 0.507 |

Table 6: Success rate of object manipulation of different skill cluster number.

| | Cubes | 1 | 2 | 3 | 4 | 5 |
|---|---|---|---|---|---|---|
| Push | **PySR** | **1.000** | **1.000** | **0.875** | **0.750** | **0.688** |
| | Neural Network | 1.000 | 0.670 | 0.613 | 0.535 | 0.417 |
| Push-Grab-Lift | **PySR** | **0.625** | **0.500** | **0.500** | **0.156** | **0.125** |
| | Neural Network | 0.502 | 0.373 | 0.367 | 0.008 | 0.003 |
| Ordered-Press | **PySR** | **0.990** | **0.938** | **0.875** | **0.813** | **0.813** |
| | Neural Network | 0.681 | 0.602 | 0.586 | 0.443 | 0.411 |

Table 7: Success rate of object manipulation using PySR and neural network.

## D.2 DETAILED RESULTS FOR MINECRAFT TASKS

The plans for different tasks generated by the MCTS algorithm are as follows:

- `Pickup-Mass-Grass`: move(workbench$_1$) → pickup(grass$_1$)

- `Pickup-Mass-Banboo`: move(workbench$_1$) → pickup(bamboo$_1$)

- `Make-Grass-Stack`: pickup(grass$_1$) → move(toolshed$_1$) → move(workbench$_1$) → make_grass_stack(workbench$_1$)

- `Make-Bamboo-Fence`: pickup(bamboo$_1$) → move(toolshed$_1$) → move(workbench$_1$) → make_bamboo_fence(workbench$_1$)

- `Make-Mass-Sticks`: move(workbench$_1$) → pickup(wood$_1$) → move(workbench$_1$) → make_stick(workbench$_1$) → pickup(wood$_2$) → pickup(wood$_3$) → pickup(wood$_4$) → pickup(wood$_5$) → move(workbench$_2$) → make_stick(workbench$_2$) → pickup(wood$_6$) → move(workbench$_1$) → make_stick(workbench$_1$) → make_stick(workbench$_1$) → make_stick(workbench$_1$) → pickup(wood$_7$) → move(workbench$_2$) → pickup(wood$_8$) → pickup(wood$_9$) → move(workbench$_1$) → make_stick(workbench$_1$) → make_stick(workbench$_1$) → make_stick(workbench$_1$) → make_stick(workbench$_1$) → pickup(wood$_{10}$) → pickup(wood$_{11}$) → pickup(wood$_{12}$) → move(workbench$_2$) → make_stick(workbench$_2$)

- `Pickup-Iron`: pickup(wood$_1$) → move(workbench$_1$) → pickup(wood$_2$) → pickup(wood$_3$) → move(toolshed$_1$) → move(workbench$_1$) → make_stick(workbench$_1$) → make_stick(workbench$_1$) → pickup(stone$_1$) → pickup(stone$_2$) → pickup(stone$_3$) → pickup(stone$_4$) → move(toolshed$_1$) → make_stone_pickaxe(toolshed$_1$) → move(workbench$_2$) → pickup(iron$_1$)

- `Multiple-Goals`: pickup(stone$_1$) → move(workbench$_1$) → move(toolshed$_1$) → pickup(stone$_2$) → pickup(wood$_1$) → pickup(stone$_3$) → pickup(stone$_4$) → pickup(stone$_5$) → move(workbench$_1$) → make_stick(workbench$_1$) → pickup(wood$_2$) → move(workbench$_1$) → make_stick(workbench$_1$) → pickup(stone$_6$) → move(toolshed$_1$) → make_stone_pickaxe(toolshed$_1$) → pickup(iron$_1$) → pickup(iron$_2$) → pickup(wood$_3$) → move(workbench$_2$) → make_scissors(workbench$_1$) → move(toolshed$_1$) → pickup(wool$_1$)

- `Make-Enhance-Table`: pickup(stone$_1$) → pickup(wood$_1$) → pickup(stone$_2$) → pickup(stone$_3$) → move(workbench$_1$) → make_stick(workbench$_1$) → pickup(wood$_2$) → pickup(wood$_3$) →

$\text{move(workbench}_1) \rightarrow \text{make\_stick(workbench}_1) \rightarrow \text{move(toolshed}_1) \rightarrow \text{make\_stone\_pickaxe(toolshed}_1) \rightarrow \text{pickup(iron}_1) \rightarrow \text{pickup(iron}_2) \rightarrow \text{move(workbench}_2) \rightarrow \text{make\_scissors(workbench}_2) \rightarrow \text{make\_paper(workbench}_2) \rightarrow \text{move(toolshed}_1) \rightarrow \text{pickup(wool}_1) \rightarrow \text{pickup(wood}_4) \rightarrow \text{move(workbench}_1) \rightarrow \text{make\_stick(workbench}_1) \rightarrow \text{pickup(stone}_4) \rightarrow \text{pickup(wood}_5) \rightarrow \text{pickup(wool}_2) \rightarrow \text{pickup(wool}_3) \rightarrow \text{pickup(wood}_6) \rightarrow \text{pickup(iron}_3) \rightarrow \text{move(workbench}_1) \rightarrow \text{make\_stick(workbench}_1) \rightarrow \text{make\_paper(workbench}_1) \rightarrow \text{pickup(iron)} \rightarrow \text{pickup(iron}_4) \rightarrow \text{move(toolshed}_1) \rightarrow \text{make\_iron\_pickaxe(toolshed}_1) \rightarrow \text{pickup(gem}_1) \rightarrow \text{move(toolshed}_1) \rightarrow \text{make\_enhance\_table(toolshed}_1)$

### D.3 DETAILED RESULTS FOR ISAACGYM TASKS

The plans for different tasks generated by the MCTS algorithm are as follows:

- `Push-n`: $\text{push}(obj_1) \rightarrow \text{push}(obj_2) \rightarrow \cdots \rightarrow \text{push}(obj_n)$.
- `Push-Grab-n`: $\text{push}(obj_1) \rightarrow \text{push}(obj_2) \rightarrow \cdots \rightarrow \text{push}(obj_n) \rightarrow \text{grab}(obj_n)$.
- `Push-Grab-Lift`: $\text{push}(obj_1) \rightarrow \text{push}(obj_2) \rightarrow \cdots \rightarrow \text{push}(obj_n) \rightarrow \text{grab}(obj_n) \rightarrow \text{lift}(obj_n)$.
- `Ordered-Press`: $\text{press}(obj^1) \rightarrow \text{press}(obj^2) \rightarrow \cdots \rightarrow \text{press}(obj^n)$.

The superscripts of `Ordered-Press` represent that press should follow the sequence.

We list some detailed results of the tasks in IsaacGym. Table 8, table 9, and table 10 demonstrate the detailed metrics of `Push`, `Push-Grab-Lift`, and `Ordered-Press`, respectively.

Additionally, we construct an experiment called `Push-Grab`. Its difficulty is between `Push` and `Push-Grab-Lift` since we expect the agent to use two skills to complete the task. The agent is required to push the cubes to their goal positions and grab one of the specified cubes. We show the detailed results under different numbers of cubes in table 11.

| Cubes | Success | Success Fraction | Color Success Fraction | Color Success | Action Success Fraction | Action Success |
|---|---|---|---|---|---|---|
| 1 | 1.000 | 1.000 | 1.000 | 1.000 | 1.000 | 1.000 |
| 2 | 1.000 | 1.000 | 1.000 | 1.000, 1.000 | 1.000 | 1.000 |
| 3 | 0.875 | 0.958 | 0.958 | 0.938, 0.938, 1.000 | 0.958 | 0.875 |
| 4 | 0.750 | 0.922 | 0.922 | 1.000, 0.875, 0.938, 0.875 | 0.922 | 0.750 |
| 5 | 0.688 | 0.900 | 0.900 | 0.938, 0.875, 0.938, 0.938, 0.813 | 0.900 | 0.688 |

Table 8: `Push`. The sequence of color success fractions follows red, green, blue, yellow, and purple.

| Cubes | Success | Success Fraction | Color Success Fraction | Color Success | Action Success Fraction | Action Success |
|---|---|---|---|---|---|---|
| 1 | 0.625 | 0.875 | 0.625 | 0.625 | 0.875 | 1.000, 0.9375, 0.688 |
| 2 | 0.500 | 0.828 | 0.719 | 0.500, 0.938 | 0.771 | 0.938, 0.875, 0.500 |
| 3 | 0.500 | 0.863 | 0.792 | 0.563, 0.938, 0.875 | 0.771 | 0.750, 0.938, 0.625 |
| 4 | 0.063 | 0.698 | 0.609 | 0.188, 0.938, 0.750, 0.563 | 0.479 | 0.438, 0.625, 0.375 |
| 5 | 0.125 | 0.643 | 0.575 | 0.3125, 0.5625, 0.5, 0.75, 0.75 | 0.438 | 0.1875, 0.625, 0.5 |

Table 9: `Push-Grab-Lift`. The sequence of color success follows red, green, blue, yellow, and purple. The sequence of action success follows push, approach, and lift.

| Cubes | Success | Success Fraction | Color Success Fraction | Color Success | Action Success Fraction | Action Success |
|---|---|---|---|---|---|---|
| 1 | 0.990 | 0.990 | 0.990 | 0.990 | 0.990 | 0.990 |
| 2 | 0.938 | 0.969 | 0.969 | 0.969, 0.969 | 0.938 | 0.938 |
| 3 | 0.875 | 0.958 | 0.958 | 0.938, 1.000, 0.938 | 0.875 | 0.875 |
| 4 | 0.813 | 0.953 | 0.975 | 0.875, 1.000, 1.000, 0.938 | 0.875 | 0.8125 |
| 5 | 0.813 | 0.950 | 0.950 | 0.875, 1.000, 1.000, 0.938, 0.938, | 0.813 | 0.875 |

Table 10: `Ordered-Press`. The sequence of color success follows red, green, blue, yellow, and purple.

## D.4 COMPOSITIONAL GENERALIZATION

For the experiments measuring compositional generalization, we provide some demonstration results in the main paper. Here, we list some of the additional results. The results are listed in Table 12 and Table 14. For the Minecraft environment, we introduce some new objects {grass, bamboo} and the corresponding crafting tasks. For IsaacGym Environment we add some new type of objects {cuboid, cylinder, star, T-block} in the environment. We can find that in most of the test cases, our model can maintain the success rate without fine-tuning the model. Also, we provide another demonstration of the experiment result in Figure 9, which is pushing the star and crafting the bamboo.

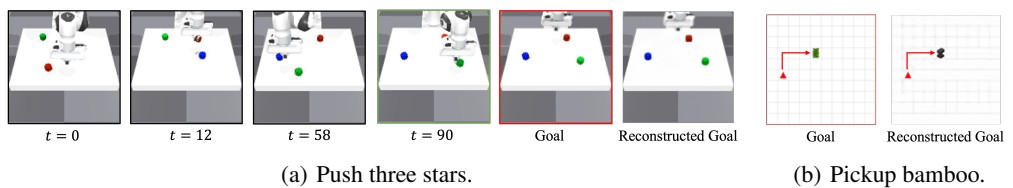

| $t = 0$ | $t = 12$ | $t = 58$ | $t = 90$ | Goal | Reconstructed Goal | Goal | Reconstructed Goal |
|---|---|---|---|---|---|---|---|

(a) Push three stars.                    (b) Pickup bamboo.

Figure 9: Compositional generalization in IsaacGym and Minecraft environments.

| Cubes | Success | Success Fraction | Color Success Fraction | Color Success | Action Success Fraction | Action Success |
|-------|---------|-----------------|------------------------|---------------|-------------------------|----------------|
| 1 | 0.938 | 0.969 | 0.938 | 0.938 | 0.969 | 1.000, 0.9375 |
| 2 | 0.938 | 0.979 | 0.969 | 0.875, 1.000 | 0.969 | 0.938, 0.938 |
| 3 | 0.563 | 0.859 | 0.813 | 0.688, 0.875, 0.875 | 0.750 | 0.688, 0.813 |
| 4 | 0.438 | 0.838 | 0.828 | 0.688, 0.875, 0.875, 0.875 | 0.656 | 0.625, 0.688 |
| 5 | 0.250 | 0.792 | 0.763 | 0.688, 1.000, 0.688, 0.688, 0.750 | 0.531 | 0.3125, 0.75 |

Table 11: `Push-Grab`. The sequence of color success follows red, green, blue, yellow, and purple. The sequence of action success fractions follows push and approach.

| Shape | Cuboid | Cylinder | Star | T-Block |
|-------|--------|----------|------|---------|
| SMORL | $0.021 \pm 0.009 / 0.070 \pm 0.008$ | $0.011 \pm 0.008 / 0.040 \pm 0.009$ | $0.030 \pm 0.011 / 0.071 \pm 0.008$ | $0.011 \pm 0.011 / 0.025 \pm 0.008$ |
| ECRL | $0.026 \pm 0.008 / 0.101 \pm 0.008$ | $0.015 \pm 0.007 / 0.066 \pm 0.017$ | $0.051 \pm 0.012 / 0.122 \pm 0.012$ | $0.013 \pm 0.009 / 0.046 \pm 0.007$ |
| GAIL | $0.000 \pm 0.000 / 0.003 \pm 0.005$ | $0.011 \pm 0.013 / 0.030 \pm 0.008$ | $0.077 \pm 0.017 / 0.153 \pm 0.010$ | $0.000 \pm 0.000 / 0.003 \pm 0.005$ |
| DiRL | $0.012 \pm 0.010 / 0.020 \pm 0.008$ | $0.018 \pm 0.007 / 0.040 \pm 0.006$ | $0.062 \pm 0.007 / 0.105 \pm 0.014$ | $0.021 \pm 0.009 / 0.055 \pm 0.016$ |
| DeepSynth | $0.113 \pm 0.009 / 0.252 \pm 0.009$ | $0.210 \pm 0.012 / 0.432 \pm 0.015$ | $0.431 \pm 0.016 / 0.629 \pm 0.005$ | $0.187 \pm 0.008 / 0.404 \pm 0.009$ |
| **Ours** | $\mathbf{0.375 \pm 0.012 / 0.750 \pm 0.005}$ | $\mathbf{0.250 \pm 0.011 / 0.729 \pm 0.005}$ | $\mathbf{0.625 \pm 0.009 / 0.833 \pm 0.006}$ | $\mathbf{0.250 \pm 0.011 / 0.667 \pm 0.009}$ |

Table 12: **Success rates and success fractions of `Push` tasks on three objects with different shapes.** To evaluate the impact of object shape on task performance, we scaled the objects by a factor of 3 along the x-axis and 1.5 along the y-axis.

### D.5 VISUALIZATION OF DLP RESULTS

We demonstrate the object reconstruction visualization of DLP in the IsaacGym and Minecraft environments.

Figure 10 and figure 11 present lists of 32 images reconstructed by DLP respectively. In the IsaacGym environment, We find that DLP focuses on objects with different colors and the gripper, while in Minecraft, object blocks and agents are clearly shown in the grid.

| Tasks | Pickup-Mass-Grass | Pickup-Mass-Bamboo | Make-Grass-Stack | Make-Bamboo-Fence |
|-------|-------------------|--------------------|--------------------|----------------------|
| SMORL | $0.270 \pm 0.011$ | $0.231 \pm 0.012$ | $0.183 \pm 0.011$ | $0.175 \pm 0.011$ |
| ECRL | $0.287 \pm 0.012$ | $0.292 \pm 0.011$ | $0.186 \pm 0.013$ | $0.179 \pm 0.011$ |
| GAIL | $0.369 \pm 0.011$ | $0.277 \pm 0.012$ | $0.267 \pm 0.010$ | $0.365 \pm 0.011$ |
| DiRL | $0.655 \pm 0.009$ | $0.563 \pm 0.010$ | $0.535 \pm 0.011$ | $0.570 \pm 0.010$ |
| DeepSynth | $0.693 \pm 0.008$ | $0.615 \pm 0.009$ | $0.651 \pm 0.009$ | $0.591 \pm 0.010$ |
| **Ours** | $\mathbf{0.969 \pm 0.004}$ | $\mathbf{0.927 \pm 0.004}$ | $\mathbf{0.865 \pm 0.008}$ | $\mathbf{0.791 \pm 0.007}$ |

Table 13: **Success rates of Minecraft tasks on new materials.** We replace wood with grass and bamboo, transforming the `Pickup-Wood` task into collecting grass and bamboo. In analogy to the `Make-Stick` task in Minecraft, the agent can then craft a grass stack using grass and construct a bamboo fence using bamboo.

| Tasks | Pickup-Iron | Make-Enhance-Table |
|---|---|---|
| Task with Figure Distortion | $0.893 \pm 0.006$ | $0.731 \pm 0.010$ |
| **Original Task** | $\mathbf{0.917 \pm 0.005}$ | $\mathbf{0.750 \pm 0.008}$ |

Table 14: **Comparison of success rates under visual distortion.** To evaluate the robustness of our pixel-based planning model, we introduce visual distortions to the material images in Minecraft and compare the resulting success rates with those obtained using the original, undistorted images.

Figure 12 and figure 13 present a comparative analysis of the original image with various transformed versions. These include images with different key points, reconstructed images, extracted foregrounds and backgrounds, and images with different types of bounding boxes. The first row depicts the original image. Key points are marked on the original image in the second row. The third row showcases the reconstructed images, which exhibit a high degree of similarity to the originals. In the fourth row, predicted key points are superimposed on the original image, with many aligning closely with objects. The fifth row highlights the top 10 key points that the agent prioritizes, which are predominantly concentrated on meaningful objects rather than empty regions. The sixth and last rows display the extracted foregrounds and backgrounds, respectively. The foreground images effectively isolate individual objects, while the backgrounds are clean and devoid of objects. The seventh and eighth rows demonstrate the application of bounding boxes to each object using two different methods: non-maximum suppression alone and non-maximum suppression in conjunction with transparency.

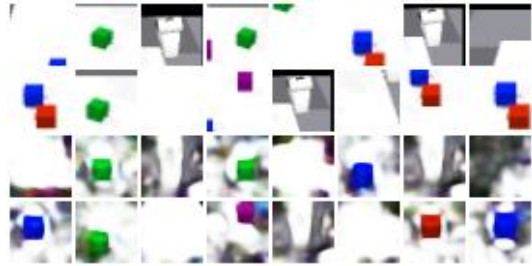

Figure 10: Object Reconstruction of DLP in IsaacGym.

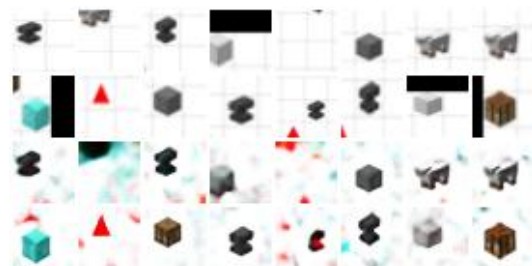

Figure 11: Object Reconstruction of DLP in Minecraft.

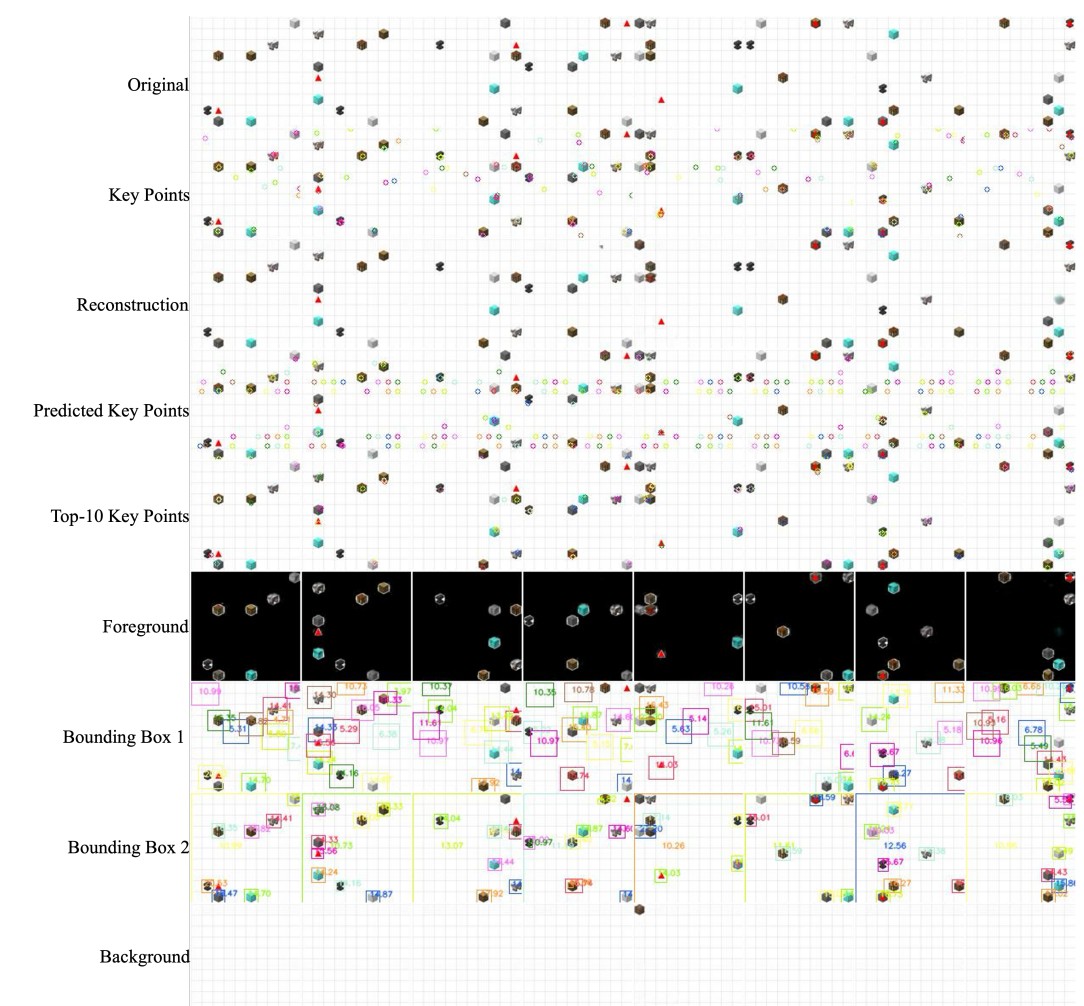

Figure 12: Visualization of DLP in IsaacGym.

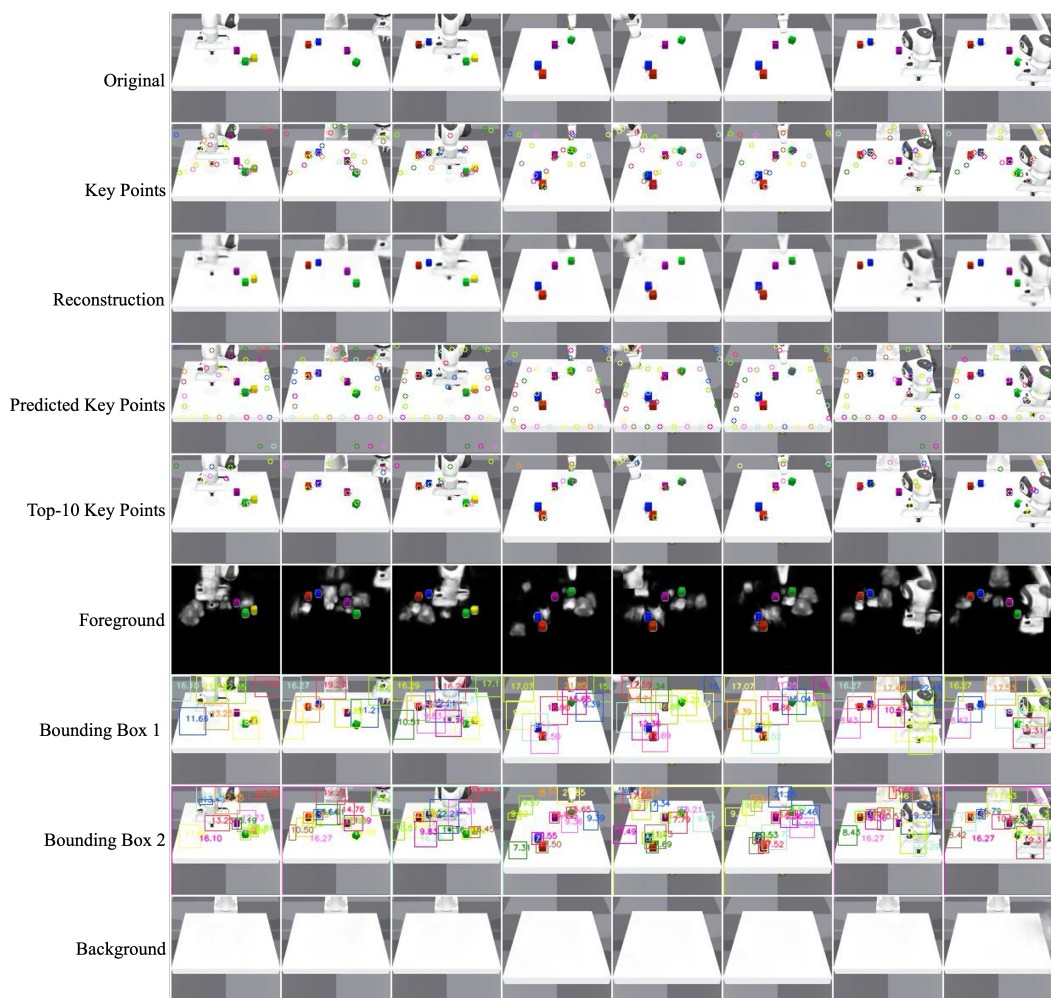

Figure 13: Visualization of DLP in Minecraft.

# E THE USE OF LARGE LANGUAGE MODELS

In the process of drafting this paper, we employed large language models (LLMs) as an auxiliary tool to enhance the quality and clarity of our written English. The primary application was to identify and correct grammatical inaccuracies, refine sentence structures, and polish academic expressions, thereby improving the overall readability and professionalism of the manuscript.

Specifically, selected paragraphs or sentences from our initial drafts were input into an LLM (e.g., DeepSeek-v3.1 or a comparable model) with explicit instructions focused solely on language checking and polishing. The prompts were designed to request grammatical corrections, suggestions for more concise or academically appropriate phrasing, and improvements in logical flow, without altering the core technical content or scientific meaning.

It is crucial to emphasize that the role of the LLM was strictly limited to that of a writing assistant. All substantive intellectual contributions, including the core ideas, theoretical framework, experimental design, data analysis, and result interpretation, remain entirely our own. The final decision to adopt any suggestion provided by the LLM was always subject to our careful review and judgment. We ensured that every change aligned with our intended meaning and adhered to the standards of academic integrity.

This use of LLMs significantly streamlined the writing and revision process, allowing us to focus more effectively on the scientific rigor and conceptual depth of our work.

