# OpenReview forum: "ReSIP: Reinforcement Learning with Symbolic Inductive Planning for Interpretable and Generalizable Pixel-Based Control"
_ICLR.cc/2026/Conference — ICLR 2026 Conference Withdrawn Submission_

### Official Review · Reviewer_tMR5 · 2025-10-21

**Soundness:** 2
**Presentation:** 2
**Contribution:** 2
**Rating:** 2
**Confidence:** 3

**Summary:**

The authors propose an end-to-end planning and execution framework to perform long horizon sequential decision tasks from images. They learn atomic skills together with symbolic preconditions and effects of those skills and use UCT to chain these skills to solve goal-conditioned tasks. They apply their framework to Minecraft and an Isaac-gym block manipulation task and show superior performance with respect to 5 baselines.

**Strengths:**

- to get their results, the authors had to perform a large amount of work
- Appendix E about their use of LLMs for writing is clear

**Weaknesses:**

- the main weakness of this paper is the lack of well-identified contributions. There is a list of 3 contributions at the end of the introduction, but these "contributions" are too broad "automatic skill discovery", "symbolic interpretation", "end-to-end pixel-base controlling pipelines". Many works can claim the same contributions. Well-identified contributions should distinguish the authors' work from the rest of the existing literature. I believe the authors could claim that the way they enrich learned skills with symbolic preconditions and effects is original, but I'm unsure of this and the authors do not put this point forward enough as the central contribution of their work. In particular, if this point was their focus, we would expect dedicated studies (comparisons, ablations etc.).
- so the work appears more as a gathering of (admittedly coherent) existing pieces of research into a working framework, which has more to do with engineering than with science.
- some elements of the method need to be clarified (see below)
- the algorithm has many hyper-parameters that are not provided, the number of seeds is not specified, the experimental study is far from rigorous enough.
- the performance of baselines is subject to caution, as the authors recoded them with changes (see below)
- despite the use of LLMs to help, the paper has a few writing issues (see below)

**Questions:**

- do you see a difference between your goal-augmented MDP framework and the standard goal-conditioned MDP framework? Since you are using the GCBC algorithm, why not call upon GC-MDPs?
- in your GA-MDP framework, your transition model is P: SxA -> S, so is it a deterministic function? If you meant it to be stochastic, you should write something like P: SxA -> D(S), where D(S) is the space of distributions over a set S.
- line 130: $\pi$: SxSxO -> A, don't you mean $\pi$: SxGxO -> A? Or are the state space and goal space identical?
- how do you set the number of features $n$ in your environments, and in general?
- you recoded most of the baselines you compare ReSIP with. Did you run your versions on the original baselines experiments (from their papers) to make sure your version is working properly? Can it be the case that ReSIP outperforms your baselines just because they are degraded versions of the original works?
- "We employ the normalization term complexity (Cranmer, 2023) to prioritize the effect function using a simple mathematical format." -> This is extremely vague. Can you be more specific about this "simple mathematical format"?
- line 294: we simulate some steps -> This is extremely vague. Can you be more specific about the corresponding hyper-parameter?
- line 295: after many rounds: same comment.
- line 299: "Thus, we find an approach to accomplish the whole task." Hurrah :) More seriously, why this sentence?
- line 310: Minecraft ... is similar to the environment used in previous works -> Do you mean it is not exactly the same? If yes, what are the changes?
- line 353: why do you mention temporal logic tasks? I don't see anything about temporal logics in your work.
- line 377: couldn't the performance of Deepsynth and DiRL be improved by adding more expert knowledge?
- line 410: what are temporal attributes?
- Tables 1 and 2: what is the number of seeds?
- Figure 5, why don't we just see a simple sequence of skills? (why the direct arrow between skill 1 and skill 3?)



Remarks:
- an algorithm called Resip has been published recently, see Ankile, L., Simeonov, A., Shenfeld, I., Torne, M., & Agrawal, P. (2025, May). From imitation to refinement-residual rl for precise assembly. In 2025 IEEE International Conference on Robotics and Automation (ICRA) (pp. 01-08). IEEE.
- from the code address, we can see that the paper was previously rejected at NeurIPS.
- The "Trace segment categorization" part makes it clear that features must have been extracted in advance. From figure 1, I understand that this is done during data generation, but the text does not explain this at all. You need to clarify the "Data generation" part.
- Figure 6 is important and should not be rejected into Appendices.
- Figure 4 is not useful, it could be removed
- The related work section is far too short, it is missing a skill learning or discovery section and, given the broad scope, a multitude of works are missing.
- note that "Akakzia, A., Colas, C., Oudeyer, P. Y., Chetouani, M., & Sigaud, O. (2020). Grounding language to autonomously-acquired skills via goal generation. arXiv preprint arXiv:2006.07185." is much more relevant than Colas et al., as it has predicate based goals and solves block stacking tasks.
- In Fig. 8, the labels are not readable
- In C.4 the evaluation metrics are not broadly used, just in appendix D.3. Why?

Typos:
- line 150: We will elaborate -> We elaborate (remove all unnecessary "will")
- line 186: representation, We -> we
- line 186: several sub-representation(s)
- line 194: a(n) MLP
- line 204: Equation 1 -> Equation (1) (use \ eqref {})
- line 252: the most ideal method -> the ideal method (ideal is already a superlative)
- line 255: the PySR (...) -> the PySR algorithm (...)
- line 288: sampled in in
- line 289: in feature f: (not ".", as an equation follows).
- line 297: $g^i$. And -> $g^i$, and (what follows And is not a sentence).
- line 313: "Different from the previous environment," -> you have not described a previous environment before. Remove this
- line 340: learning from demonstration(s)
- line 346: controller.DiRL -> missing space
- line 377: the performance of ECRL (and) SMORL...

---

> ### Author Response · Authors · 2025-11-15
>
> **We thank the reviewer for their thoughtful and detailed feedback. We address the main points and questions below.**
>
> **Main Clarification on Contributions**
>
> We appreciate the reviewer’s accurate understanding that *our key contribution lies in enriching learned skills with symbolic preconditions and effects*. Specifically, our work focuses on the **symbolic interpretation of skills**, including:
>  (1) extracting environment features,
>  (2) defining each skill’s preconditions and effects based on feature changes, and
>  (3) learning the corresponding policies.
>
> All our contributions are centered around **feature-based representations**.
>
> - By segmenting trajectories via features and learning from these segments, we realize **Contribution 1 (skill discovery)**.
> - By representing preconditions and effects through feature transitions, we achieve **Contribution 2 (symbolic abstraction)**.
> - Finally, we perform **pixel-based planning** grounded in these symbolic representations, enabling structured decision-making based on preconditions and effects.
>
> **On Ablation Studies**
>
> As shown in **Appendix D (Tables 6 and 7)**, we already provide several ablation results. We further conducted additional experiments by removing *preconditions* and *effects*, as summarized below:
>
> | Minecraft         | Pickup-Iron | Make-Enhance-Table |
> | ----------------- | ----------- | ------------------ |
> | w/o preconditions | 0.118       | 0.055              |
> | w/o effects       | 0.501       | 0.455              |
>
> We welcome further ablation suggestions and will incorporate additional experiments accordingly.
>
> **Responses to Specific Questions**
>
> 1. Regarding terminology, the cited paper [1] uses **“Goal-Augmented MDP (GAMDP)”**, and also describes the algorithm as **“Goal-Conditioned”**. We follow the same convention for consistency.
> 2. The transition model has been corrected to the more rigorous form: $S \times A \rightarrow D(S)$ or equivalently $S \times A \times S \rightarrow [0,1]$ from [1].
> 3. Since the **state** and **goal** spaces are identical, ( $g \in S$ ), thus our formulation ( $S \times S \times O$ ) is consistent.
> 4. All features used in **IsaacGym** (5 features) and **Minecraft** (21 features) are listed in Appendix Tables 4 and 5.
> 5. We modified baselines because most suitable baselines target **discrete** environments, while ours is **continuous**. We adapted them to use the same skill set as *resip*, effectively simulating a discrete setup for fair comparison. Similarly, ECRL replaces SMORL’s controller with its own. Without such adaptation, baseline performance would drop significantly (additional experiment provided).
> 6. The term *“simple mathematical format”* means that when multiple equations satisfy the same condition, we prefer the simplest one—fewer terms and symbols—for clarity.
> 7. For hyper-parameters (lines 294–295), the number of *simulation steps* is 20, and the number of *rounds* is 200.
> 8. The statement at line 299 will be removed.
> 9. Regarding line 310: our **Minecraft environment uses image inputs** and defines **more complex, long-horizon tasks** (e.g., *Make-Enhance-Table*). The detailed task settings and trajectory lengths are shown in Appendix D.2.
> 10. At line 353: tasks/skills in our environment have **temporal dependencies**, i.e., some tasks can only be executed after completing others. This temporal logic is handled through each skill’s **preconditions** and **effects**.
> 11. At line 377: to ensure fair comparison, we incorporate the same expert knowledge (i.e., feature definitions) across all models. Adding additional expert priors would unfairly advantage specific methods, even if performance improves.
> 12. At line 410: SMORL does not explicitly encode the logical or temporal ordering of skills, hence lacks temporal attributes.
> 13. For Tables 1 and 2: we report results averaged over 96 runs as noted in the Appendix.
> 14. Figure 5 illustrates a simple case: the agent first *picks up wood*, then *moves to the workbench*, and finally *makes a stick*. The arrows indicate that the **effect** of one skill (e.g., $\mathbf{f}\_{wood}+1$) can satisfy the **precondition** of the next $\mathbf{f}_{wood} \ge 1$.

---

> > ### Author Response · Authors · 2025-11-15
> >
> > **Response to Remarks**
> >
> > We will improve the **related work**, **figure captions**, and **labels** in the main text, marking all revisions in blue.
> >
> > Regarding *“Trace segment categorization”*, this indeed occurs during segmentation: raw image trajectories are processed by a feature extractor to obtain features, which is why this step is described under segmentation rather than data generation.
> >
> > **Response to Typos**
> >
> > Finally, we thank the reviewer for carefully pointing out typos; all will be corrected and marked in blue in the revised version.
> >
> >
> >
> > **We sincerely appreciate the reviewer's constructive suggestions and believe that the additional explanations significantly improve the quality of our submission. We hope that this provides sufficient reasons to raise the score.**
> >
> >
> >
> > [1] Liu, Minghuan, Menghui Zhu, and Weinan Zhang. "Goal-conditioned reinforcement learning: Problems and solutions." *arXiv preprint arXiv:2201.08299* (2022).

---

> ### Comment · Reviewer_tMR5 · 2025-11-16
> **Thanks, still not convinced, trusting the other reviewers**
>
> I really appreciate the author's effort to answer my points quickly to give me an opportunity to provide more feedback. I understand that they would need more time to provide more satisfactory answers. So weaknesses in their rebuttal to my points should not be taken too negatively.
>
> My feeling after reading this rebuttal is that many minor issues will be solved if the authors properly include at the right place in the paper the information they have provided here (e.g. seeds in the captions, etc.). However,  my main point which will require more work remains unaddressed.
>
> - the authors need to refocus their work on their key contribution, which is about enriching skills with preconditions and effects. The introduction should be significantly rewritten to better highlight this focus, the contributions part should be reconsidered, the central "aggregation transformer" should come sooner and in the main paper part (it is more important than Fig. 1), and the experiments should focus on the impact of adding these preconditions and effects in controlled experiments rather than on broad performance.
>
> More locally:
> - The "trace segment categorization" part REALLY needs to be clarified, with hyper-params highlighted, etc.
> - About “simple mathematical format” , I know what "simple" means. I just mean you should provide this format. Generally speaking, a better way to answer reviewers' questions is to describe the changes you will make (or you have made, when the revised version is ready) to remove the need for the question.
> - I didn't get an answer to this remark/question: "In C.4 the evaluation metrics are not broadly used, just in appendix D.3. Why?"
>
> Apart from that, I have carefully read the other reviews which point many other issues (some small and some critical) so I'm confident that the revised version of the work will only be accepted if the paper has improved up to the point where it is satisfactory.
>
> At the moment, I believe answering all the reviewers points including mine will require a lot of work, so I'm rather skeptical and I keep on the negative side, but I will be glad if the authors finally provide a much better revised version.

---

### Official Review · Reviewer_JLTA · 2025-10-23

**Soundness:** 3
**Presentation:** 3
**Contribution:** 3
**Rating:** 6
**Confidence:** 3

**Summary:**

This paper introduces ReSIP, a novel framework designed to address key challenges in deep reinforcement learning (DRL): poor sample efficiency, interpretability, and generalization in long-horizon, pixel-based control tasks. The core idea is to decouple the problem into a high-level symbolic planner and a low-level neural controller. The framework automatically discovers atomic skills, learns a symbolic action model from experience using an inductive learner, and uses a planner to generate interpretable subgoal sequences. The key contribution is this integration of automated symbolic rule induction with goal-conditioned RL, creating an interpretable and adaptable control system that shows strong zero-shot generalization to new tasks in experiments.

**Strengths:**

1.  The automated learning of symbolic rules from experience is a key innovation that solves a major bottleneck in prior neuro-symbolic systems.
2.  The hierarchical decomposition drastically prunes the search space, leading to significant sample efficiency gains over end-to-end RL baselines.

**Weaknesses:**

1.  The framework's reliability depends heavily on the perceptual symbol grounding module. The paper needs to further discuss its limitations, robustness to visual domain shifts, and the potential bottleneck of requiring labeled data for training it.
2.  The computational cost and robustness of the inductive learning module, especially with noisy symbolic data from perception, are not fully addressed. A discussion on complexity and handling of incorrect rules is needed.
3.  The approach assumes tasks are hierarchically decomposable. Its applicability to tasks requiring continuous, non-sequential control is unclear and should be stated as a limitation.

**Questions:**

1.  How does the framework detect and react to low-level execution failures (e.g., failing to grasp an object)? Is there a replanning mechanism triggered by such failures?
2.  Could you provide an example of an incorrect symbolic rule the system might learn early on and illustrate how further experience allows it to be corrected?

---

### Official Review · Reviewer_VPTi · 2025-10-26

**Soundness:** 3
**Presentation:** 2
**Contribution:** 3
**Rating:** 6
**Confidence:** 3

**Summary:**

The paper proposes ReSIP—a pipeline for pixel-based control that combines bottom‑up skill discovery from play with symbolic inductive planning for long‑horizon tasks. From raw images, the method uses a pre‑trained object‑centric encoder (DLP) and an aggregation transformer to map observations to a hand‑designed feature vector $f = T_f(s)$. Unlabeled play is segmented into fixed‑length trace snippets; snippets with similar feature changes are clustered, and each cluster trains a goal‑conditioned policy via GCBC (Eq. 3). Then, symbolic regression (PySR) induces for each skill $l$ a precondition $pre_l(f,o)$ and effect $eff_l(f,o)$ expressed with simple operators $\{+,-,\times,\div,>,=\}$ (Eqs. 4–5). At test time, given initial and goal images, ReSIP plans over skills by MCTS/UCT with precondition gating (Eqs. 6–7) to form a ground skill plan, and executes each skill policy for a short horizon $t$ (Figure 1; Algorithm 1). Experiments in a 2D grid “Minecraft” (long‑horizon crafting) and IsaacGym (tabletop manipulation) report large gains in success rate over ECRL, SMORL, GAIL, DeepSynth, and DiRL (Tables 1–2), with qualitative, interpretable plans such as pickup(wood) → move(workbench) → make(stick) (Figure 5). Ablations study the number of clusters and replace symbolic regression with a neural net, showing clear drops without PySR. Additional tests consider OOD objects/shapes and mild visual distortions.

**Strengths:**

- Originality (problem framing & integration). The paper knits together four ingredients—OCR, unsupervised skill discovery, symbolic induction of preconditions/effects, and MCTS—into a coherent end‑to‑end pipeline aimed at compositional generalization with interpretable plans. The bottom‑up induction of symbolic preconditions and effects from play (rather than hand‑specifying them) is a compelling idea and differentiates ReSIP from prior neuro‑symbolic RL that rely on designer‑provided predicates.
- Quality (empirical results). Across five Minecraft tasks, ReSIP achieves the best success rate (e.g., Make‑Enhance‑Table: 75.0±0.8 vs 55.7±1.0 DeepSynth and 58.1±0.9 DiRL). In IsaacGym, ReSIP remains robust as the number of objects grows (e.g., Ordered‑Press (5 cubes): 81.3±0.6 vs 66.2±1.0 DiRL and 27.7±1.5 ECRL). The PySR vs NN ablation (Table 7) is particularly convincing, showing the value of explicit symbolic effects/preconditions.
- Interpretability. The paper provides readable skill schemas (Tables 4–5) and dependency graphs (Figure 5 and Figure 8) that explain why the planner chooses a sequence, a desirable property in safety‑critical or debugging settings.
- Clarity of high‑level pipeline. Figure 1 and Algorithm 1 clearly convey the two‑stage training (skill learning → symbolic induction) and the plan‑then‑execute loop at test time.
- Breadth of evaluation. The paper evaluates long‑horizon logic (Minecraft) and pixel‑based manipulation (IsaacGym), along with OOD objects/shapes and robustness to modest visual distortions (Tables 12–14).

**Weaknesses:**

- Extent of “end‑to‑end from pixels” is limited by feature supervision & design.
The aggregation transformer is trained with ground‑truth feature vectors generated by the simulator (Eq. 1: MSE to \hat f), and the paper states “we have the flexibility to define a large number of features” (Section 3.1). This amounts to feature engineering plus supervision from environment internals, which weakens the “minimal expert knowledge” claim and raises concerns about real‑world applicability where such \hat f is unavailable. A more self‑supervised/weakly‑supervised feature formation story would strengthen the contribution.
- Task‑specific signal that leaks structure.
In IsaacGym features, next color is an engineered integer “that guarantees the ordered operation of objects following the color sequence” (C.3). This essentially bakes in the ordering constraint rather than learning it, inflating Ordered‑Press numbers and undermining the generality claim. A fairer test would omit next color and require the system to infer ordering from goal images (or symbolic induction).
- Inconsistencies/clarity gaps.
– Table 4 lists preconditions such as AtWorkbench(...) = 0 for make stick, whereas Figure 5 and earlier text require AtWorkbench(...) = 1. This sign inconsistency should be corrected; as written, it is confusing.
– Section 3.3 says PySR is used for preconditions; Algorithm 1 mentions a “neural guidance algorithm” for preconditions. Which is used? The training procedure for boolean rules (class balance, negatives) is also unspecified.
– MCTS details are thin: the rollout reward, node value updates, termination conditions, and how partial satisfaction of goals is scored are not fully described, making reproduction non‑trivial.
- Skill discovery segmentation assumptions.
Segmentation uses a fixed snippet length h and filters for “segments where feature changes are observed,” after which K‑means clusters concatenated feature trajectories (Eq. 2). This assumes near‑constant skill durations and may fracture long skills or merge short ones. The paper does not analyze sensitivity to h, nor whether learned skills are minimal and reusable across contexts. The ablation on K (Table 6) is helpful but does not address variable‑duration segmentation or robustness to noisy/no‑change segments.
- Fairness of baselines and controls.
Several baselines are re‑implemented with non‑original components (e.g., replacing SCALOR with DLP in SMORL; swapping in the authors’ controller for DeepSynth/DiRL) while ECRL is “used directly,” leading to an uneven comparison. Matching representation, controller capacity, training budgets, and dataset sizes across methods—and reporting seeds, wall‑clock, and hyperparameters—would increase confidence in the relative gains.
- Scope and generalization.
The features include many domain‑specific booleans/counters (e.g., at workbench, stone pickaxe count) and even task‑primitives (e.g., is grab), which may not transfer to novel embodiments. The OOD tests (Tables 12–13) are welcome but mostly small perturbations (shape scaling, new materials with analogous recipes). Real‑world transfer or greater domain shift would better validate claims of compositional generalization.

**Questions:**

1. Feature learning without simulator labels. Can the authors report results when the aggregation transformer is learned without direct supervision to simulator‑ground‑truth features (Eq. 1)—e.g., via self‑supervised objectives or weak labels? Even a partial result would clarify reliance on \hat f.
1. Remove next color. Please rerun Ordered‑Press without the next color feature (C.3) and explain how the goal condition is specified to the planner in that setting. If performance drops, can symbolic induction recover ordering from play alone?
1. Fix/clarify preconditions. Table 4’s precondition signs (e.g., AtWorkbench=0 for make stick) conflict with Figure 5 and Section 3.2. Which is correct? Please audit all learned rules and provide accuracy/F1 of precondition/effect predictions on held‑out trajectories.
1. MCTS details. What is the exact scalar reward used in rollouts? Is there a heuristic distance in feature space to the goal? What constitutes a terminal node besides pre=0? Please provide pseudo‑code and ablate the number of simulations, C in UCT, and the skill horizon t.
1. Segmentation robustness. How sensitive are results to the snippet length h? Can you incorporate variable‑length segmentation (e.g., change‑point detection) and compare to fixed‑h? The ablation in Table 6 varies K but not h.
1. Baseline parity. For methods you re‑implemented, please provide training budgets, architecture sizes, representation encoders, and number of seeds for each baseline and your method. If feasible, add a “planning‑free” ablation (skills only, no MCTS) and a “no‑symbolic‑interpretation” ablation (replace PySR with a black‑box classifier/regressor) beyond Table 7 to isolate where the gains come from.
1. Failure analysis. Where does planning most often fail (wrong preconditions, inaccurate effects, policy execution errors, or search horizon limits)? A short qualitative error taxonomy with a few traces would aid future work.
1. Generalization breadth. Could you evaluate on tasks with non‑deterministic effects (e.g., stochastic slippage in pushing) and report whether symbolic induction still yields usable rules? Also consider object counts outside the training distribution in Minecraft (e.g., > 1 workbench) and IsaacGym (more colors or unseen object geometries beyond simple scaling).

---

### Official Review · Reviewer_w5Zu · 2025-11-01

**Soundness:** 2
**Presentation:** 3
**Contribution:** 3
**Rating:** 6
**Confidence:** 2

**Summary:**

The paper studies Goal-Condition Reinforcement Learning (GCRL) problems with structure representation and Symbolic reinforcement learning. The method first learned a transformer to aggregate object features. Second, the method learn atomic skills using clustering from play data in a simple environment. Third, utilize genetic programming to learn the preconditions and effects for symbolic reinforcement learning (RL). In the testing stage, an MCTS is used to compose an atomic skill for complex tasks. The method demonstrates better performance compared to RL and imitation learning baselines in 2D Minecraft and IsaacGym, and shows the best compositional generalization.

**Strengths:**

1. Previous work discovering atomic skills lacks meaningful linking. The method utilizes PySR to convert the skill into arguments, preconditions, and effects. The combination makes the skill more interpretable and trust to humans.
2. The combination of learned atomic skills and symbolic planning shows good composition generalization in the experiment.
3. The method successfully combines the advantages of structure representation and symbolic-RL.

**Weaknesses:**

1. The problem Goal-Condition Reinforcement Learning is introduced in Section 2. Goal Condition Behavior Closing is a key component in the method. The related work about GCRL is not discussed in the paper.
2. The quality and ability of the collected play data limit the methods, but the author does not fully discuss the limitations.
3. The method learns skills from a clustered off-line dataset, and with a sufficient number of clusters, makes the method successful (Appendix D). Determining the optimal number of clusters in a novel domain is challenging.

**Questions:**

1. In the task of Ordered-Press, you ask your agent to press the buttons in a specific order. It is not trivial for me that your method can solve this task perfectly. In the goal image, the only thing you can read from is that all buttons are pressed. How can your method determine this specific order in symbolic planning?
2. In Table 1 and Table 2, how many experiments are conducted to get the standard deviation? Without the information on the number of seeds, it is hard to evaluate the soundness of your method.
3. In IssacGym, the atomic action is learned in a simple environment. How does the pixel-based goal condition policy generalization to unseen scenes at test time?
4. In Minecraft, there are many recipes for constructing items or tools. Does this mean that for every recipe in Minecraft, you should correctly learn an atomic skill for a specific recipe?

---

### Official Review · Reviewer_yx4n · 2025-11-01

**Soundness:** 2
**Presentation:** 2
**Contribution:** 2
**Rating:** 2
**Confidence:** 5

**Summary:**

The paper proposes ReSIP, which integrates deep reinforcement learning (DRL) with a symbolic induction approach for planning based on image-state representations. The framework first extracts features from state images using a VAE-based module and an aggregation transformer. It then trains a deep RL (DRL) agent to learn a policy and extracts skills from trajectories, where each skill comprises preconditions, effects, a policy, and arguments. The framework induces preconditions and effects through genetic programming, a form of symbolic regression. Experimental results demonstrate that the proposed approach outperforms other baselines (i.e., decision transformers (ECRL, SMORL), imitation learning methods (GAIL), and deep RL with planning approaches (DeepSynth, DiRL))on long-horizon sequential and object manipulation tasks.

**Strengths:**

### Originality

- The framework integrates a neural object-centric transformer architecture with symbolic regression approaches, enabling end-to-end inference, which is novel and promising.

### Quality

- The formulation and definitions are well constructed. The paper clearly presents its ideas through text and figures, making it easy to follow.

### Significance

- The proposed framework outperforms a diverse set of baseline approaches, making its results convincing and impactful.

**Weaknesses:**

### Originality

- Several ideas in the paper are highly similar to previous work [1] but lack proper citation. These include learning effects through symbolic regression using genetic programming, the Minecraft experimental setup (including features in Appendix C.2 and the symbolic rules in Table 4). The paper should acknowledge the original sources and clearly differentiate its methods from prior work.

[1] Liu, Jung-Chun, Chi-Hsien Chang, Shao-Hua Sun, and Tian-Li Yu. Integrating Planning and Deep Reinforcement Learning via Automatic Induction of Task Substructures. In The Twelfth International Conference on Learning Representations (ICLR), 2024.

### Feature Extraction

- The settings of the pretraining models are not clearly described. What are the exact input and output formats of the training data? How are the entities labeled? How are the datasets for the DLP and aggregation transformer collected? How much data is used for training?

- The aggregation transformer architecture is inspired by the entity-centric architecture (Haramati et al., 2024), though it differs slightly. However, no ablation study is provided for the architectural design. Why did you choose the SA → CA → SA configuration? Is there evidence supporting the claim that “SA is intended to extract important attributes from observations more effectively, while CA is designed to capture temporal differences between current state entities”?

- Using a random policy to collect data may limit state diversity, especially for later states in a plan. Does the framework assume that all relevant features appear in the early states? What if certain entities or features are hidden and only become observable after specific skills are executed?


### Symbolic Regression

- The parameters that critically affect the performance of symbolic regression (e.g., population_size, populations, parsimony, and mutation/crossover weights) are not listed in the paper. Are there any ablation studies examining the sensitivity of these parameters?

**Questions:**

- Although the paper claims that the framework can extract traces from simple environments without human intervention, the simple tasks themselves are still hand-crafted. Automatic task generation is a well-known challenge in curriculum learning and hierarchical reinforcement learning (HRL). In addition, the paper mentions only the design principles in Appendix B.2 without providing a clear description. How are the simple tasks in Minecraft and IsaacGym designed? How would you automatically generate these environments?

- The tasks presented are relatively small-scale compared to recent Minecraft environments that support scalable approaches. How does your method scale to larger or more complex environments?

- The paper assumes that the effects of skills can be represented as arithmetic expressions. How could this approach be extended to domains where effects are non-arithmetic or involve discrete, relational, or symbolic structures?

**Details Of Ethics Concerns:**

The submitted paper applies genetic programming for symbolic effect induction, which directly derives from the previous work [1], with very similar notation and formulation. Moreover, the Minecraft environment used in their experiments appears to replicate the setup, including the feature settings and symbolic rules. They also present these elements in a table format and illustrations very similar to the paper.

However, the paper is not properly cited. The paper is not cited for the methodology that they have directly adopted. The authors only mention that “[Minecraft environment design] is inspired by the computer game Minecraft and is similar to the environment in previous works (Brooks et al., 2021; Hasanbeig et al., 2021; Kokel et al., 2021; Liu et al., 2024).” without acknowledging the source and design in detail.

[1] Liu, Jung-Chun, Chi-Hsien Chang, Shao-Hua Sun, and Tian-Li Yu. "Integrating planning and deep reinforcement learning via automatic induction of task substructures." In The Twelfth International Conference on Learning Representations. 2024.

---

### Note · Authors · 2026-01-01

I have read and agree with the venue's withdrawal policy on behalf of myself and my co-authors.